# Remodeling of gene regulatory networks underlying thermogenic stimuli-induced adipose beiging

Seoyeon Lee[1], Abigail M. Benvie[1], Hui Gyu Park[2], Roman Spektor [3], Blaine Harlan[3], J. Thomas Brenna[1,2], Daniel C. Berry [1] & Paul D. Soloway [1,4✉]

Beige adipocytes are induced by cold temperatures or β3-adrenergic receptor (Adrb3) agonists. They create heat through glucose and fatty acid (FA) oxidation, conferring metabolic benefits. The distinct and shared mechanisms by which these treatments induce beiging are unknown. Here, we perform single-nucleus assay for transposase-accessible chromatin sequencing (snATAC-seq) on adipose tissue from mice exposed to cold or an Adrb3 agonist to identify cellular and chromatin accessibility dynamics during beiging. Both stimuli induce chromatin remodeling that influence vascularization and inflammation in adipose. Beige adipocytes from cold-exposed mice have increased accessibility at genes regulating glycolytic processes, whereas Adrb3 activation increases cAMP responses. While both thermogenic stimuli increase accessibility at genes regulating thermogenesis, lipogenesis, and beige adipocyte development, the kinetics and magnitudes of the changes are distinct for the stimuli. Accessibility changes at lipogenic genes are linked to functional changes in lipid composition of adipose. Both stimuli tend to decrease the proportion of palmitic acids, a saturated FA in adipose. However, Adrb3 activation increases the proportion of monounsaturated FAs, whereas cold increases the proportion of polyunsaturated FAs. These findings reveal common and distinct mechanisms of cold and Adrb3 induced beige adipocyte biogenesis, and identify unique functional consequences of manipulating these pathways in vivo.

[1] Division of Nutritional Sciences, College of Agriculture and Life Sciences, Cornell University, Ithaca, New York, NY, USA. [2] Dell Pediatric Research Institute, Departments of Chemistry, Pediatrics, and Nutrition, Dell Medical School and the College of Natural Sciences, University of Texas at Austin, Austin, TX, USA. [3] Field of Genetics, Genomics, and Development, Department of Molecular Biology and Genetics, Cornell University, Ithaca, New York, NY, USA. [4] Department of Biomedical Sciences, College of Veterinary Medicine, Cornell University, Ithaca, New York, NY, USA. ✉email: soloway@cornell.edu

Adipose tissues have a central role in maintaining systemic energy homeostasis[1,2]. Specifically, white adipose tissue (WAT) is considered a key regulator of energy storage and distribution and regulates a myriad of local and distal metabolic responses[3]. In contrast, reducing the environmental temperature can recruit and activate brown adipose tissue (BAT) to dissipate stored energy as heat *via* non-shivering thermogenesis[1,2]. To accomplish thermogenesis, brown adipocytes contain specialized mitochondria that express a high level of uncoupling protein 1 (UCP1), which can disconnect the electron transport chain by collapsing the proton gradient, generating heat rather than chemical energy[1,2]. In addition to brown adipocytes, cold temperature exposure can produce brown-like fat cells within WAT depots[4]. These so-called beige or brite adipocytes resemble brown fat cells in that they consume glucose and fatty acids (FAs) to perform thermogenesis[1]. In addition to cold temperatures, several other stimuli such as β3-adrenergic receptor (Adrb3) agonists, and peroxisome proliferator activated receptor gamma (Pparg) agonists have been shown to promote beige adipocyte biogenesis[5–7]. Regardless of the beige adipocyte inducer, it appears that the activation of thermogenesis plays a critical role in shifting energy expenditure and mobilization of glucose and lipids from the blood. Additionally, identification and activation of human BAT have been linked to averting diet-induced weight gain and insulin resistance[2]. Thus, defining the cellular and transcriptional regulators of beige adipocyte formation under various beige fat-inducers could aid in developing targeted therapies to counteract metabolic disease.

Cold exposure stimulates the sympathetic nervous system to release norepinephrine, which activates adipose beiging through Ardb3[1]. Due to the importance of Adrb3 signaling, Adrb3 agonists are often interchangeably used with cold exposure to mimic cold stimulation of BAT[8,9]. Although the two stimuli share major pathway features, distinct effects of cold exposure and Adrb3 agonists on beiging have been reported[6]. Lineage tracing studies, genetic necessity tests, and pharmacological strategies have suggested that cold temperatures stimulate WAT resident perivascular adipocyte progenitor cells (APCs) to undergo *de novo* adipocyte formation[6,10]. However, compared to cold, it has been reported that Adrb3 activation stimulates pre-existing white adipocytes to convert into beige adipocytes[6]. The mechanisms underlying the emergence of beige adipocytes remain incompletely characterized. It is also unknown whether these mechanisms differ when cold or Adrb3 activation induces beiging. Moreover, of clinical relevance, the use of Adrb3 agonists is contraindicated due to off-target side-effects such as tachycardia and hypertension[11]. Yet recent efforts have identified a more selective Adrb3 agonist, mirabegron, which is clinically used to treat overactive bladder conditions, demonstrating clinical promise to improve metabolism in obese humans by stimulating beige fat formation but without the cardiometabolic effect[12]. However, beyond the understanding of different cellular sources to generate beige fat much remains about the cellular, molecular, and metabolic differences between cold temperature- and Adrb3-induced beige fat. Critically, understanding these differences would be essential for enhancing clinical care to foster metabolic fitness and promote white fat loss.

The resolving power of single-cell technologies have enabled the characterization of cellular heterogeneity, and their transcriptional and epigenomic states in complex tissues[13–15]. Recently, single-cell RNA-sequencing (scRNA-seq) and single-nucleus RNA-sequencing (snRNA-seq) were applied to characterize cell populations in mature adipocytes and/or the stromal vascular fraction (SVF) of adipose tissue[8,9,16–19]. While these studies have yielded tremendous insight into the transcriptomes of white and beige adipose tissues, only a few studies have attempted to discern differences between cold exposure and Adrb3 activation[18,20]. Revealing gene regulatory networks (GRNs) is of significant interest as the interplay of *trans*-acting regulatory factors and *cis*-acting regulatory elements likely control the emergence of beige adipocytes and their metabolic programs. Yet, despite the importance of chromatin states, adipose tissues are a poorly represented node in chromatin state datasets.

Here, we performed single-nucleus assay for transposase-accessible chromatin with high-throughput sequencing (snATAC-seq) on WAT collected across the beige adipogenic time course induced by cold exposure or CL-316,243 (CL), an Adrb3 agonist. We characterized GRNs for the beige adipogenic process and developmental trajectory of cellular changes accompanying adipose tissue beiging. A comparative analysis for commonality and heterogeneity of two thermogenic agents showed that cold temperatures increase gene accessibility regulating glycolytic processes, whereas Adrb3 activation increases cAMP responses. Moreover, while we observed changes in thermogenic, lipogenic, and beige fat development genes in response to cold and Adrb3 activation, these responses had unique kinetics and magnitudes. Interestingly, we observed changes in lipogenic genes that could be attributed to functional changes in WAT lipid composition. Both cold and CL increased FA elongation, but Adrb3 activation more induced the formation of monounsaturated FAs (MUFAs). Taken together, our analysis provides insights into mechanisms underlying beiging, and the shared and distinct effects of cold and Adrb3 activation on WAT.

## Results

**snATAC-seq reveals distinct cell types in inguinal adipose tissue.** In order to understand the dynamic chromatin remodeling during adipose tissue beiging, we performed snATAC-seq of inguinal WAT (iWAT) from mice exposed to cold temperature (C; 6 °C) or treated with CL at room temperature (RT; 22 °C) for 0, 1, 3, or 7 days (Fig. 1a). Histological assessment showed both cold exposure and CL induced beige adipocyte formation in iWAT depots (Supplementary Fig. 1a). Nuclei from whole adipose tissues were used to obtain cell types in entire iWAT to overcome technical bias related to isolation of cells or SVF. A total of 32,552 cells passed quality control from seven groups (day0RT, day1C, day3C, day7C, day1CL, day3CL, and day7CL) and three individual mice for each group (Supplementary Fig. 2a–s). To identify the cell types, we performed dimensionality reduction and unsupervised clustering based on top 50% variable peaks using Signac[21,22]. This revealed 12 highly consistent cell clusters, visualized using Uniform Manifold Approximation and Projection (UMAP) (Fig. 1b). We annotated clusters using differentially accessible genes, gene ontology (GO) analysis, cell type specific marker genes, transcription factor (TF) motif enrichment, and GREAT analysis (Fig. 1c and Supplementary Fig. 3a–e). We further confirmed that the annotation is consistent with the cell type identities predicted by publicly available adipose tissue scRNA-seq and snRNA-seq data with high correlation rates (Supplementary Fig. 3f, g). Adipocytes constitute about 90% of adipose tissue volume but comprise only 20% of total cells[23]. In agreement with our expectation, we found an average of 22% of adipocytes in our snATAC-seq dataset (Supplementary Fig. 4a, b). The average proportion of other non-adipocyte cell types in our dataset was about 72% for immune cells, 3% for APCs, and 2% for stromal cells (Supplementary Fig. 4a).

**Dynamic changes in adipocytes and identification of thermogenic beige adipocytes.** We analyzed the changes in cell abundance in response to cold and CL treatment in each cell type (Fig. 1d). However, the differential abundance testing of discrete

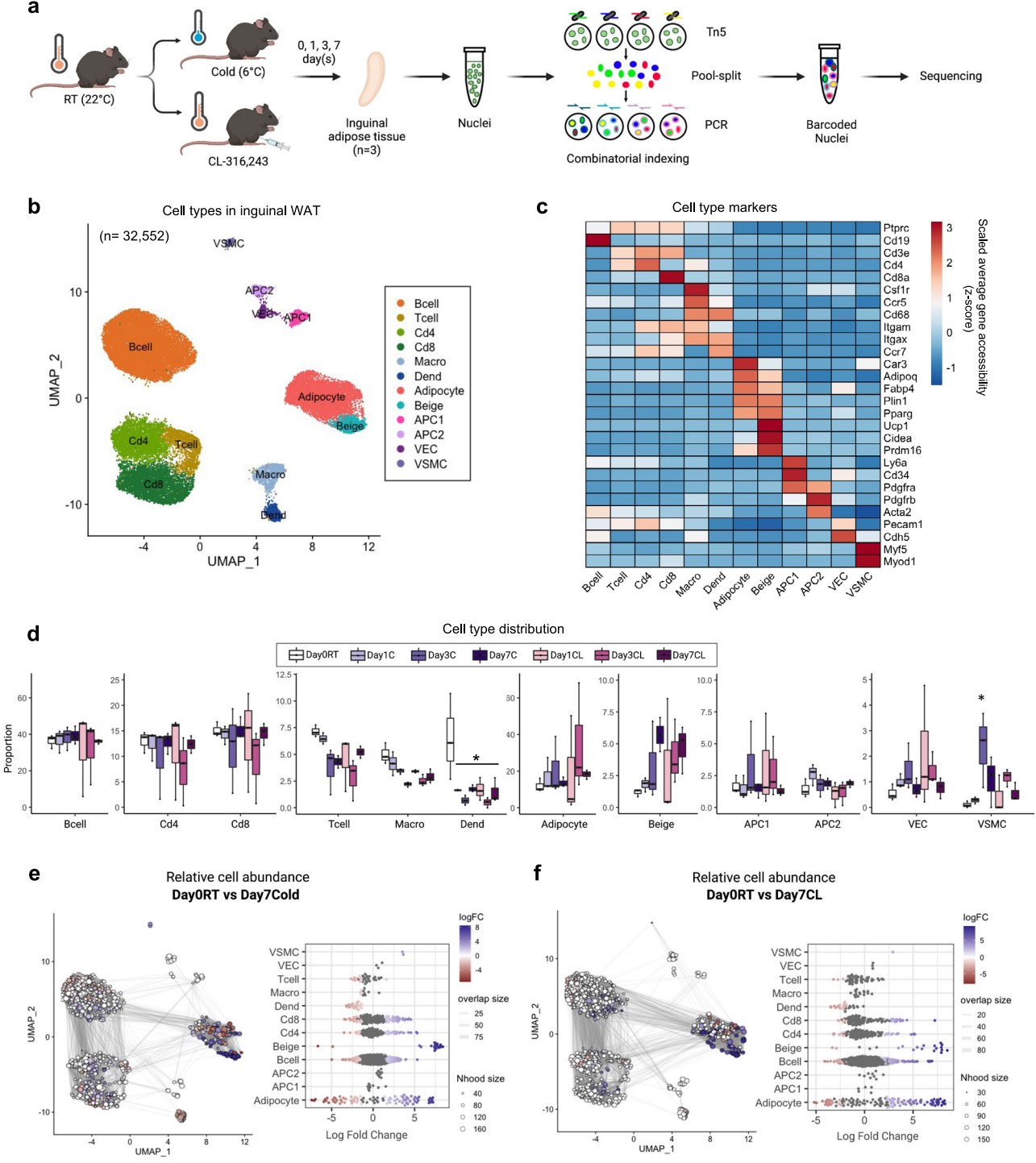

**Fig. 1 Cell type identification in inguinal white adipose tissue. a** Schematic outline of the workflow. Two-month-old male C57BL/6 mice were exposed to cold (6 °C) or CL-316,243 (CL; 1 mg/kg/mouse/day) for 1, 3, or 7 days ($n = 3$ per group). Cartoons of mice and thermometers created with BioRender.com. Inguinal adipose tissues collected at day 0, 1, 3, and 7 after exposure was subjected to snATAC-seq. **b** UMAP plot of 32,552 cells that passed quality control. Colors represent 12 different clusters determined by unbiased clustering. **c** Heatmap of normalized accessibility of cell-type marker genes. **d** Relative proportions of each cell type per group ($n = 3$ per group). *Adjusted $p$-value<0.05, ANOVA multiple comparisons test with Bonferroni's post-hoc test was performed. **e**, **f** Milo analysis of cell neighborhood abundance changes in Day 0 RT *vs.* Day 7 cold **e**, or Day 7 CL **f** on UMAP plots. Size of points indicates the number of cells in a neighborhood; lines represent the number of cells shared between adjacent neighborhoods. Points are neighborhoods (Nhood), colored by the log fold differences at the two time points (FDR 10%). Beeswarm plots show the distribution of the log-fold in defined clusters. Macro Macrophage, Dend Dendritic cell, APC1 Adipocyte progenitor cell 1, APC2 Adipocyte progenitor cell 2, VEC Vascular endothelial cell, VSMC Vascular smooth muscle cell.

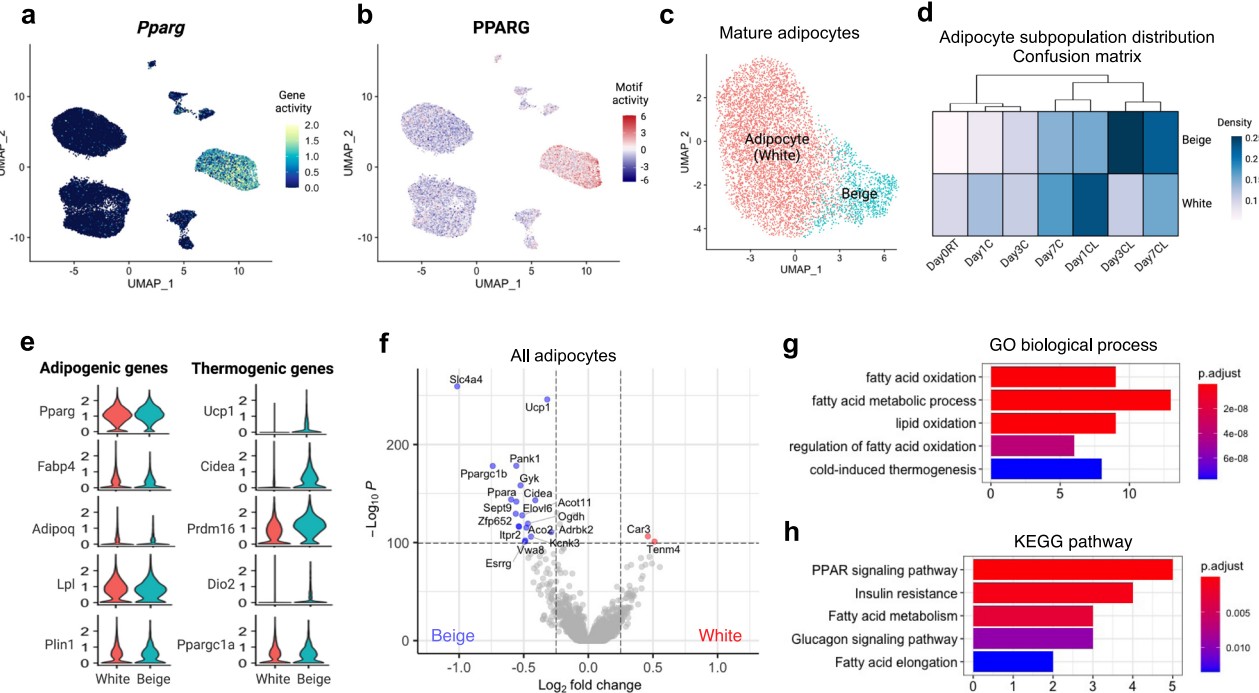

**Fig. 2 Identification of beige adipocytes within mature adipocytes. a** UMAP plot of normalized *Pparg* gene accessibility. **b** UMAP plot of normalized PPARG motif enrichment. **c** UMAP plot of white and beige adipocytes among mature adipocytes ($n = 7{,}004$). **d** Heatmap of the relative cell abundance in white and beige adipocyte clusters for each group. **e** Normalized accessibilities of adipogenic or common adipocyte genes and thermogenic or beige-selective genes. **f** Differentially accessible genes between white and beige adipocytes. Positive and negative log2-fold changes respectively indicate increased accessibility in white and beige adipocytes. Genes with adjusted *p*-value $< 10^{-100}$ & Abs(log2 fold-change) > 0.25 are colored and labeled. **g**, **h** GO **g** and KEGG pathway **h** analysis of genes enriched in beige adipocytes compared to white adipocytes. X-axis indicates number of genes. Color indicates adjusted *p*-value.

clusters relies on predefined clusters and limits the ability to detect shifts in cell states within clusters. Thus, to further analyze the abundance changes caused by cold and CL treatment without relying on discrete clusters, we used Milo[24] to assign cells to neighborhoods having similar cellular states. The neighborhoods around adipocytes showed the highest variation in the log2 fold changes of their UMAP coordinates after cold and CL treatment. This indicated, that adipocytes underwent especially dynamic changes in their responses to cold and CL relative to other cell types (Fig. 1e, f and Supplementary Fig. 4c–h).

Mature adipocyte populations have high accessibility at the *Pparg* gene and differentially accessible peaks in these cells are enriched for PPARG motifs (Fig. 2a, b). Two distinct adipocyte populations were distinctively separated from other populations on the UMAP and identified as white or adipocytes and beige (Figs. 1b and 2c). The proportion of beige adipocytes gradually increased from 1% to 5% with increased exposure time to both cold or CL treatments (Figs. 1d and 2d). Among neighborhoods identified by Milo[24], those around beige adipocytes were increased after cold and CL treatment compared to day 0 at RT, confirming accumulation of beige adipocytes (Fig. 1e, f and Supplementary Fig. 4c–h).

General adipocyte marker genes were highly accessible in both adipocyte populations, while beige/brown adipocyte marker genes were highly accessible in beige adipocyte population (Fig. 2e). The top five genes enriched in white adipocytes were *Car3, Lrp3, Tenm4, Trabd2d*, and *Cmklr1* (Fig. 2f); whereas the top five genes enriched in beige adipocytes were *Slc4a4, Ucp1, Pank1, Ppargc1b*, and *Gyk* (Fig. 2f). GO analysis revealed that the genes enriched in beige adipocytes are related to FA oxidation and thermogenesis (Fig. 2g). KEGG pathway analysis showed that the genes enriched in beige adipocytes are also associated with PPAR signaling

pathway and FA metabolism (Fig. 2h). Taken together, the cluster we identified as beige adipocytes showed clear thermogenic signatures compared to white adipocytes.

**Beige-specific *cis*- and *trans*-acting gene regulatory networks.** Chromatin accessibility analysis reveals key GRNs including TFs and *cis*-regulatory elements underlying dynamic genetic and epigenetic programs. To comprehensively characterize GRNs activated in beige adipocytes, we identified genomic regions with higher accessibility in beige adipocytes, which we refer to as beige-specific peaks. We firstly performed GREAT analysis[25] to predict biological functions of the beige-specific peaks using annotations of the nearby genes. Predicted functions of the beige-specific regions were related to brown fat cell differentiation and FA oxidation (Fig. 3a), providing additional confidence in our original identification of beige adipocytes. We further extended these analyses by focusing on distal beige-specific peaks not residing in annotated genes. Using cicero[26], we analyzed co-accessible peaks and identify putative *cis*-regulatory interactions and the genomes that are more likely to be enhancers of the linked genes. For instance, a set of beige-specific peaks in the upstream of *Ucp1* showed high co-accessibility with each other and *Ucp1* promoter (Fig. 3b). Similarly, other sets of co-accessible beige-specific peaks were linked to genes including *Ppara, Pdk4, Ppargc1b, Acot11, Arhgef37, Slc4a4, Col27a1, Dio2, Elovl6, Kcnk3*, and *Ppargc1a*. These genes themselves exhibited high accessibility in beige adipocytes (Supplementary Fig. 5a). If intergenic beige-specific peaks include functional enhancers that are active in beige adipocytes, we expected to find H3K27ac marks associated with the beige-specific peaks. To make this determination, we referred published H3K27ac ChIP-seq data collected from white and beige adipocytes[27]. Over 6,000 beige-specific peaks overlapped with

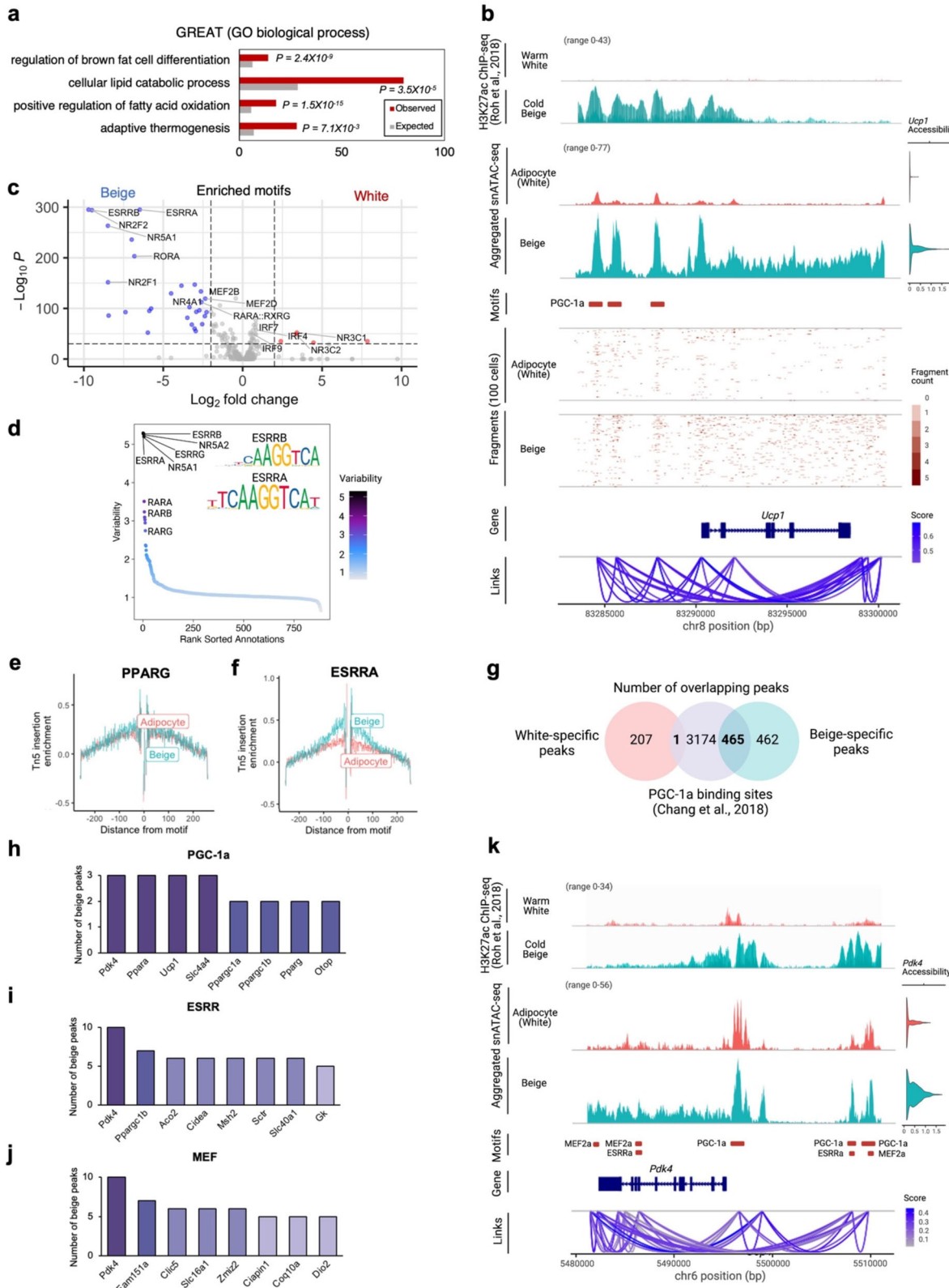

regions carrying H3K27ac (Supplementary Fig. 5b). These overlapping regions included peaks upstream of *Ucp1* that were co-accessible with the *Ucp1* promoter in beige adipocytes that were largely absent from white adipocytes (Fig. 3b). This indicates that the beige-specific peaks are likely to include a large number of functional enhancers active in beige cells.

To gain further understanding of the GRNs in beige adipocytes, we contrasted TF binding motifs enriched in beige-specific, *vs* white-specific peaks. Motifs enriched in beige adipocytes included members of the ESRR, MEF2, ROR, NR2, and NR4 families (Fig. 3c and Supplementary Fig. 5c, d). It was previously reported that peroxisome proliferator-activated

**Fig. 3 Beige-specific transcription factor activity and chromatin interaction networks. a** Biological function of beige-specific peaks by GREAT analysis. **b** Genome tracks showing the aggregated snATAC-seq profiles near the *Ucp1* locus in adipocytes. H3K27ac ChIP-seq signal tracks (*n* = 2 for each)[27] show the aggregated H3K27ac signals in white and beige adipocytes. snATAC-seq signal tracks show the aggregated snATAC-seq signals in white and beige adipocytes. Red boxes indicate beige-specific regions overlapping with PGC-1α binding sites[39]. Fragment coverage show the single cell profile (each row) of randomly selected 100 single cells from white and beige adipocytes respectively. Links show *cis*-coaccessibility network with multiple connections between peaks around *Ucp1*. **c** Motif enrichment for white and beige adipocytes. Positive and negative log2-fold changes respectively indicate enrichment in white and beige adipocytes. Motifs with adjusted *p*-value < $10^{-30}$ & Abs(log2 fold-change) > 2.0 are colored and labeled. **d** Most variable TF motifs are ranked. Motif sequence logos for the top variable motifs are shown. **e, f** TF footprints for PPARG **e** and ESRRA **f** from (white) adipocytes and beige adipocytes. Tn5 DNA sequence bias was normalized by subtracting the Tn5 bias from the footprinting signal. X-axis is distance to motif center (bp). **g** Venn diagram showing the number of overlaps between white-specific or beige-specific peaks and PGC-1α binding sites[39]. **h–j** The number of overlapping beige-specific peaks with PGC-1α binding sites[39] **h**, ESRRs **i**, and MEFs **j** motifs from *cisbp* database[41]. The genes closest to the overlapping peaks are sorted by the number of peaks. **k** Genome tracks showing the aggregate snATAC-seq profiles near the *Pdk4* locus in adipocytes. H3K27ac ChIP-seq signal tracks (*n* = 2 merged)[27] show the aggregated H3K27ac signals from white and beige adipocytes. snATAC-seq signal tracks show the aggregated snATAC-seq signals in white and beige adipocytes. Red boxes indicate beige-specific regions overlapping with PGC-1α[39], ESRRα[42], and MEF2α[43] ChIP-seq binding sites. Links show *cis*-coaccessibility network with multiple connections between peaks around *Pdk4*.

receptor coactivator-1α (PGC-1α), a key regulator for brown adipose, promoter has two MEF2-binding sites and its transcriptional activation is mediated by MEF2[28]. Especially, ESRRα and ESRRβ were the most highly variable motifs in adipocytes (Fig. 3d). Whereas the footprint pattern of the general adipose regulator PPARg was similar between white and beige adipocytes (Fig. 3e), beige adipocytes showed a distinct TF footprint for ESRRα (Fig. 3f). Loss of ESRR isoforms was shown to reduce thermogenic capacity of BAT upon adrenergic stimulation[29]. These results collectively highlight specific regulation of ESRR and its importance in beige adipocytes. In addition to ESRR and MEF2, NR2 and NR4 family members can be functionally important to induce beiging. Especially, NR4A family members were shown to bind to a regulatory region of *Ucp1* and enhance *Ucp1* expression in mouse and human adipocytes[30]. On the other hand, the most enriched TF motifs in white adipocytes is NR3C family that serves as glucocorticoid receptors (GR) (Fig. 3c). GR can function as a TF and bind to glucocorticoid response elements. It was reported that excess glucocorticoids reduce the thermogenic activity of BAT and inhibit the expression of *Ucp1*, suggesting that glucocorticoids are negative regulators of BAT thermogenesis[31–34]. In accordance with this, GR ChIP-seq data showed that NR3C1 and NR3C2 were enriched during re-warming of beige adipocytes at thermoneutrality (30 °C), suggesting that NR3Cs are involved in whitening of beige adipocytes[27]. Collectively, these results demonstrate that our snATAC-seq data revealed positive and negative transcriptional regulators of beige adipocyte development.

PGC-1α is a key transcriptional coactivator of white-to-brown transition and regulates pathways including mitochondrial FA oxidation[35] and mitochondrial biogenesis[28,36–38]. To get better insight into how PGC-1α functionally relates to beige-specific peaks we identified, we intersected the beige-specific peaks with PGC-1α binding sites which were obtained from publicly available PGC-1α ChIP-seq data from BAT[39]. We found that 50%, or 465 beige-specific peaks overlapped with PGC-1α binding sites, while only 1 white-specific peaks overlapped with PGC-1α binding sites (Fig. 3g and Supplementary Fig. 5e). Notably, *Ucp1* upstream has three beige-specific peaks which are PGC-1α binding sites, and overlapped with H3K27ac marks (Fig. 3b). In addition, *Pdk4, Slc4a4,* and *Ppara* also have three beige-specific peaks that have binding motifs for PGC-1α (Fig. 3h). These genes themselves exhibited high accessibility and expression level in beige adipocytes (Supplementary Fig. 5f–h). These results strongly suggest that PGC-1α, in concert with the TF motifs enriched in beige-specific peaks identified in the present study play important roles in beige adipocyte function.

To identify potential target genes of beige-specific TFs and their coactivator PGC-1α, we identified the genes satisfying four criteria: (1) Tight linkage to beige-specific peaks, (2) having binding motifs for TFs enriched in beige adipocytes obtained from JASPAR2020[40] or *cisbp* database[41], (3) have validated binding sites for ESRRα[42], MEF2α[43], and PGC-1α[39] based on published ChIP-seq data, (4) overlap with H3K27ac[27] marks in beige adipocytes. Among genes satisfying all criteria, *Pdk4* was identified as a target gene for beige-specific TFs and their coactivator (Fig. 3h–j and Supplementary Fig. 5i). The binding sites for PGC-1α, ESRRs, MEF2s, NR2s, and NR4s were all enriched at beige-specific peaks at the locus; accordingly, *Pdk4* is a nexus for their coordinated regulation (Fig. 3h–k and Supplementary Fig. 5i). PDK4 enzyme inactivates the pyruvate dehydrogenase complex (PDH), resulting in a decrease in the formation of acetyl-CoA from glucose and a metabolic shift from glucose oxidation to FA oxidation[44]. The recent loss-of-function study revealed that FA oxidation was blunted in *Pdk4* knockdown human adipose-derived stem cells[45]. Taken together, these data reveal the GRNs including TFs and *cis*-regulatory elements and their potential target genes underlying beiging process.

**Transcription factor modules associated with beige adipocyte developmental trajectories.** Our finding that the TFs function coordinately at beige-specific peaks motivated us to identify TF modules for beige adipocyte development. Three TF modules were revealed by unsupervised hierarchical clustering based on the similarity of TF activity computed by chromVAR[46] across all individual adipocytes (Fig. 4a). Module 1 contained NR3C, TEAD, and STAT family motifs. It was previously shown that knockdown of TEAD4 increased expression of PPARg and C/EBPα, master regulators for adipogenesis[47]. Also, adipocyte specific STAT1 knockout promoted PGC-1α expression and enhanced mitochondrial function in WAT depots[48]. Therefore, the module 1 may represent negative regulators in beige adipogenic process. Module 3 consisted of ROR, NR2, NR4, ESRR, and MEF family motifs, which were enriched in beige adipocytes. The module 3 transcriptional regulators play critical roles in mitochondrial biogenesis and thermogenesis by interacting with PGC-1α in beige adipocytes[28,36–38]. Lastly, module 2 contains C/EBP, RXR, PPAR, and NFI family motifs. RXRs play an important role in *Ucp1* induction in brown adipocytes[49]. RXRs can form heterodimers with PPARs[50]. PPARa regulates adipogenesis and FA oxidation[51]. C/EBPs function with PPARg to promote adipocyte development[52]. Also, it was reported that NFIA binds to brown fat-specific enhancers prior to brown-fat cell differentiation along with C/EBPβ in WAT and BAT[53]. The motifs in the module 2 have relatively higher correlations with both module 1 and module 3, suggesting intermediate states or shared features between white and beige adipocytes for adipocyte development.

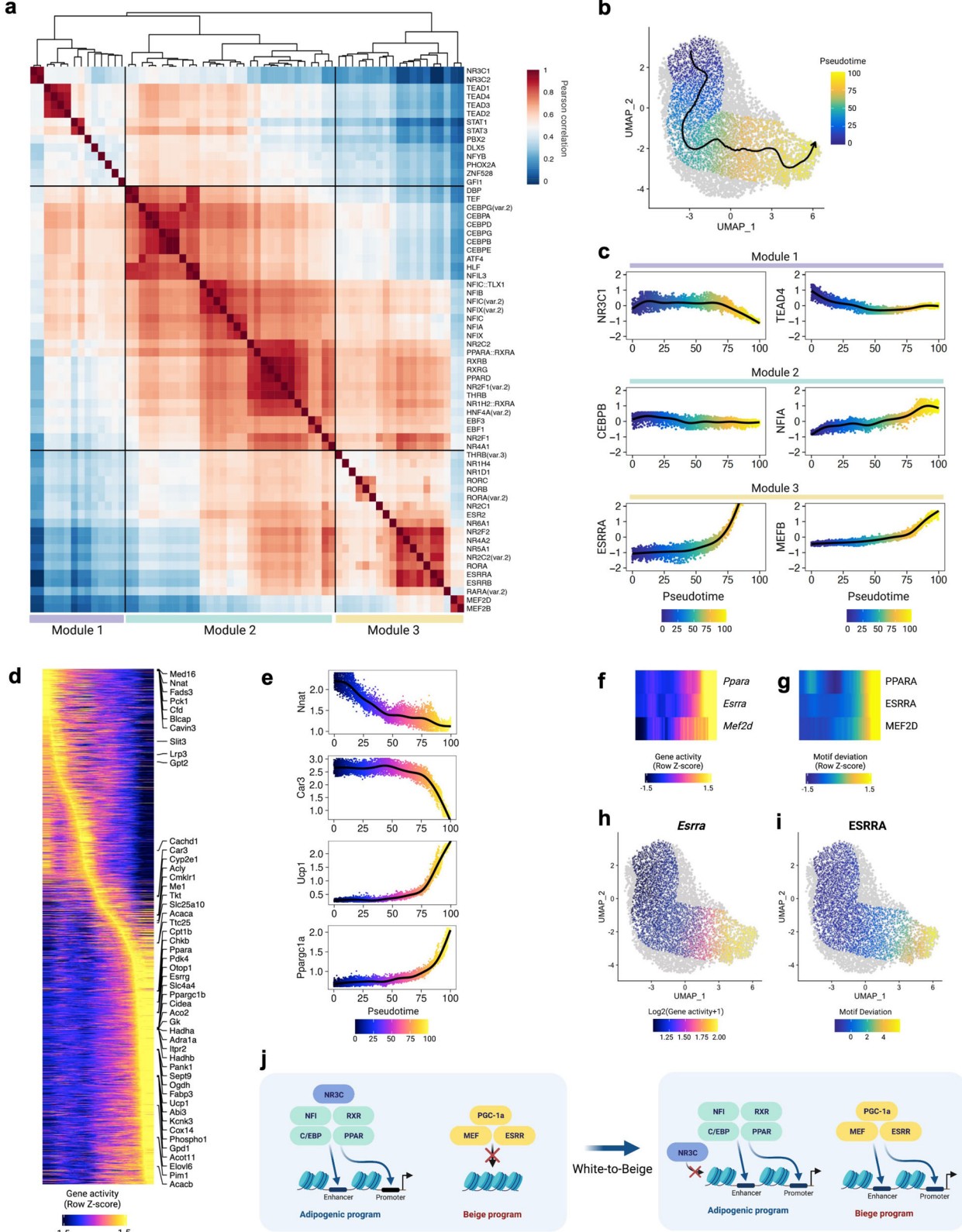

Given these results, and the facts that (1) individual cells can move between cell states in the same cell type, and (2) cells may exist in transition between states[54], we sought to find a path covering the heterogeneous cellular states in the context of continuous beige adipocyte developmental trajectory. We constructed pseudotime trajectory of adipocytes. The inferred trajectory passes from one point where most of day 0 cells are

located and to another where most of day 7 cells are located on UMAP (Fig. 4b). In line with our TF module analysis, the activity of TF motifs in module 3 was enriched in the later stage of trajectory, whereas the activity of TF motifs in module 1 was depleted (Fig. 4c). The activity of C/EBPβ and NFIA in module 2 were not significantly changed or gradually increased respectively (Fig. 4c). We ordered the genes based on changes in gene

**Fig. 4 Uncovering transcription factor modules and beige adipocyte developmental trajectory. a** Hierarchical clustering of top variable TF motifs (chromVAR variance >= 1.8) based on their activity scores across individual adipocytes. **b** Adipocyte trajectory. The smoothed arrow represents pseudotime trajectory in the UMAP embedding. **c** Motif enrichment dynamics along adipocyte pseudotime trajectory. **d** Heatmap of gene accessibility along adipocyte pseudotime trajectory. Top variable genes are labeled on the right side of the heatmap. **e** Gene accessibility dynamics along adipocyte pseudotime trajectory. **f, g** Side-by-side heatmaps of gene accessibility scores **f** and motif deviation scores **g** for TFs, for which the inferred gene accessibility is positively correlated with the chromVAR TF deviation across adipocyte pseudotime trajectory. **h, i** Gene accessibility of *Esrra* **h** and motif deviation of ESRRA (**i**) on the UMAP embedding. **j**, Model for the chromatin state dynamics at TF binding motifs through beige adipocyte development. NR3C in TF module 1 might bind to white adipocyte-selective promoters or enhancers to maintain chromatin states of white adipocytes. NFI, RXR, PPAR, and C/EBP in TF module 2 might promote adipocyte development in both white and beige adipocytes. In beige adipocyte, MEF and ESRR in TF module 3 with PGC-1α might promote transcription of beige-selective genes in a coordinated manner.

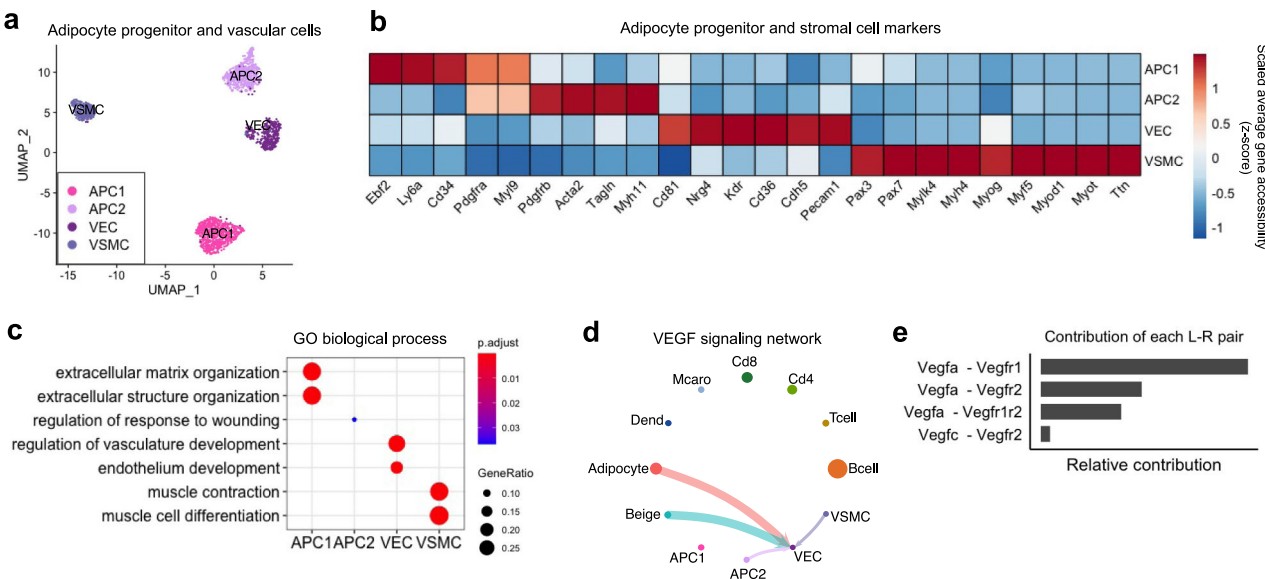

**Fig. 5 Cell–cell interaction for vasculature development. a** UMAP plot of adipocyte progenitor and vascular endothelial and smooth muscle cells (*n* = 1,861). **b** Heatmap of normalized gene accessibility for cell-type markers. **c** GO analysis using the top 100 differentially accessible genes for each cell type. **d** Circle plot displaying VEGF signaling network inferred by gene accessibility of ligand-receptor pairs. Circle sizes are proportional to the size of each group. The circle and line colors represent the senders. The arrows indicate the receivers that receive signal from the corresponding senders. Thicker line indicates a stronger signal interaction. **e** Major contributors of ligand-receptor pairs of VEGF signaling network among the senders and the receiver. APC1 Adipocyte progenitor cell 1, APC2 Adipocyte progenitor cell 2, VEC Vascular endothelial cell, VSMC Vascular smooth muscle cell.

accessibility over pseudotime trajectory (Fig. 4d and Supplementary Fig. 6a, b). It showed that adipocytes lose gene accessibility of *Med16, Nnat, Car3,* and *Fads3* at early stage (Fig. 4d, e), and there is a gradual gain of accessibility for *Acly, Me1,* and *Acaca* at the intermediate stage (Fig. 4d). As we expected, *Ucp1* was highly accessible at the later stage (Fig. 4d, e). Additionally, *Otop1, Cidea, Ppargc1b, Elovl6, Slc4a4, Ppara,* and *Pank1* were highly accessible in later pseudotime (Fig. 4d).

Finally, we identified positively correlated genes and TFs across pseudotime during beige adipocyte development. The cells at the later developmental stage gained accessibility at *Ppara, Esrra,* and *Mef2d* genes and their corresponding TF motifs (Fig. 4f, g). Specifically, the cells with high *Esrra* gene accessibility also had high ESRRA motif accessibility (Fig. 4h, i). Collectively, these emerging patterns of TF-encoding gene and TF motif accessibility across the adipocyte trajectory define mechanisms underlying environmentally-induced and transcriptionally-controlled adipocyte beiging (Fig. 4j).

**Adipose progenitors and vascular network expansion in inguinal adipose tissue**. Beige adipocytes can arise *via* transdifferentiation of unilocular mature adipocytes[4,55] but also by *de novo* differentiation from various progenitors[6,10,56]. In our snATAC-seq data, two subtypes of APCs were identified (Fig. 5a).

*Pdgfra*, a marker for APCs, was accessible in both APC clusters (Fig. 5b). APC1 was uniquely marked by accessibility at *Ebf2, Cd34,* and *Ly6a* (Sca1), whereas APC2 had higher accessibility at *Pdgfrb* and mural cell marker genes including *Acta2* and *Myh11* (Fig. 5b).

APCs reside in a vascular niche, and thus, the development of APCs is closely related to the vasculature[6,10,57]. In our snATAC-seq data, two stromal cell types were identified, vascular endothelial cells (VECs) and vascular smooth muscle cells (VSMCs). VECs were uniquely marked by accessibility at endothelial cell marker genes such as *Pecam1* and *Chd5*, while VSMCs had higher accessibility at smooth muscle cell marker genes including *Mylk4, Myh4, Myog, Myf5,* and *Myod1* (Fig. 5b). We found that the abundance of VECs and VSMCs tended to increase following cold and CL treatment (Fig. 1d). Since they are two essential cell types in blood vessels, the expansion of VEC and VSMC populations is closely associated with vessel growth. In line with this, the genes enriched in VECs were related to vasculature development (Fig. 5c).

To gain insight into how the expansion of vasculature is regulated in iWAT, we used ligand-receptor pairs to infer cellular communication[58]. This analysis revealed that adipocytes are likely to interact with VECs through vascular endothelial growth factor (VEGF) signaling, mainly through *Vegfra* and *Vegfr1*

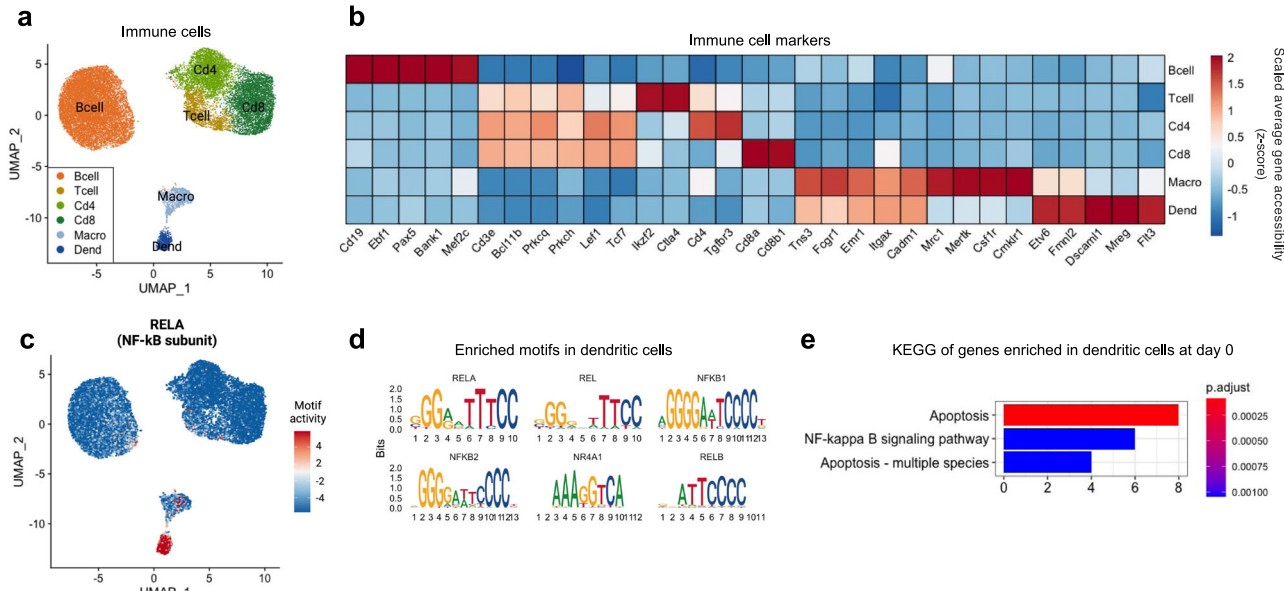

**Fig. 6 Alterations in pro-inflammatory immune cell compartment. a** UMAP of immune cell populations (*n* = 23,687). **b** Heatmap of scaled gene accessibility for cell-type markers. **c** UMAP of RELA motif enrichment. **d** Motif sequences enriched in dendritic cell population. **e** GO analysis of genes enriched in dendritic cells at day 0 RT. X-axis indicates number of genes. Color indicates adjusted *p*-value. Macro Macrophage, Dend Dendritic cell.

(Fig. 5d, e). VEGF is a major growth factor that promotes the proliferation of endothelial cells and the growth of new blood vessel in adipose tissue[59]. WAT vascular expansion is important for formation and maintenance of APC niche[7]. These results suggest that vascular cells and adipocytes in iWAT form a reciprocal feedback loop to promote angiogenesis in a paracrine manner and expansion of vascular cells provides a niche for APCs.

**Decreased abundance of inflammatory immune cells after cold and CL treatment.** Immune cells are another critical constituent of the adipose microenvironment that influence adipose tissue function and metabolic homeostasis[60]. In our snATAC-seq data, we found lymphocytes, including B cells, CD4+ and CD8+ T cells, and myeloid cells, including macrophages and dendritic cells (Fig. 6a, b). We observed a significant decrease in cell abundance of myeloid origin, especially dendritic cells after cold and CL treatment (Fig. 1d). Compared to other immune cells, the motifs for NF-kB family, including RELA, REL, NF-kB1, and NF-kB2, were highly enriched in dendritic cells (Fig. 6c, d). NF-kB is a key mediator of cytokine signaling and NF-kB signaling activates transcription of various proinflammatory genes[61]. Consistently, GO analysis showed that the less accessible genes after cold and CL treatment in dendritic cells are associated with NF-kB signaling pathway (Fig. 6e). These results suggest that dendritic cells in adipose tissue contribute to proinflammatory microenvironment, mainly through NF-kB signaling, and thus, a decrease in dendritic cell population prevents activation of inflammatory responses after cold exposure and CL treatment.

**Shared and distinct mechanisms of cold exposure and CL responses in adipocytes.** It has been suggested that beige adipocytes produced by different thermogenic stimuli may have unique genetic signatures and different metabolic properties[6]. Therefore, we examined similarities and differences of activated gene programs resulting from cold and CL treatments. We first identified genes that were more accessible after 1, 3, and 7 days of cold or CL treatment compared to day 0 (Fig. 7a and Supplementary Fig. 7a–i). We next identified those genes that had

elevated accessibility in both treatment groups, or in just one treatment group at the 3 or 7 day time points in order to identify shared and distinct mechanisms (Fig. 7b and Supplementary Fig. 7j–l). GO analysis showed that the genes enriched after both cold and CL treatment were associated with ribose phosphate metabolic process, FA and acyl-CoA metabolic process, and adaptive thermogenesis (Fig. 7c). The genes more enriched after cold exposure were involved in glycolytic process (Fig. 7d); whereas the genes enriched in CL were related to GTPase activity and cAMP signaling (Fig. 7e). These findings are consistent with the facts that Adrb3 activation, like other G protein-coupled receptor signaling cascades, triggers cAMP signaling pathways; and in the case of Adrb3 activation, this leads to induction of thermogenesis[62].

The genes that are commonly more accessible by cold and CL treatment for 1, 3, and 7 days were involved in FA metabolism, specifically *de novo* lipogenesis (*DNL*) (Fig. 7a). Interestingly, we found some kinetic differences in accessibility of lipogenic genes between cold and CL across time course (Fig. 7f). In cold, the accessibility of lipogenic genes gradually increased from day 1 to day 7, whereas the accessibility of lipogenic genes was highly increased at day 3 in CL. In contrast, the accessibility of glycolytic genes gradually increased after cold exposure, while no significant changes were observed after CL treatment (Fig. 7g). Therefore, cold and CL share common pathways during beiging, yet differ in specific signaling cascades and in timing of changes in gene accessibility.

**Cold and CL alter the composition of lipid classes in inguinal adipose tissue.** The genes encoding enzymes involved in *DNL* and FA elongation such as *Acaca*, *Acly*, *Fasn*, and *Elovl6* underwent profound changes in adipocytes after cold exposure and CL treatment (Fig. 7a and Supplementary Fig. 8a, b). FA synthetase (FAS), encoded by *Fasn*, converts malonyl-CoA into palmitic acid, a major product of *DNL*[63]. Then, FAs can be further elongated by FA elongases (ELOVLs)[63]. If any of these changes were functionally important, we expect that FAs extracted from adipose tissue would exhibit different profiles from RT after cold or CL treatment, or that profiles found in cold and CL may differ as well. Therefore, we measured a total of 21 lipid species from

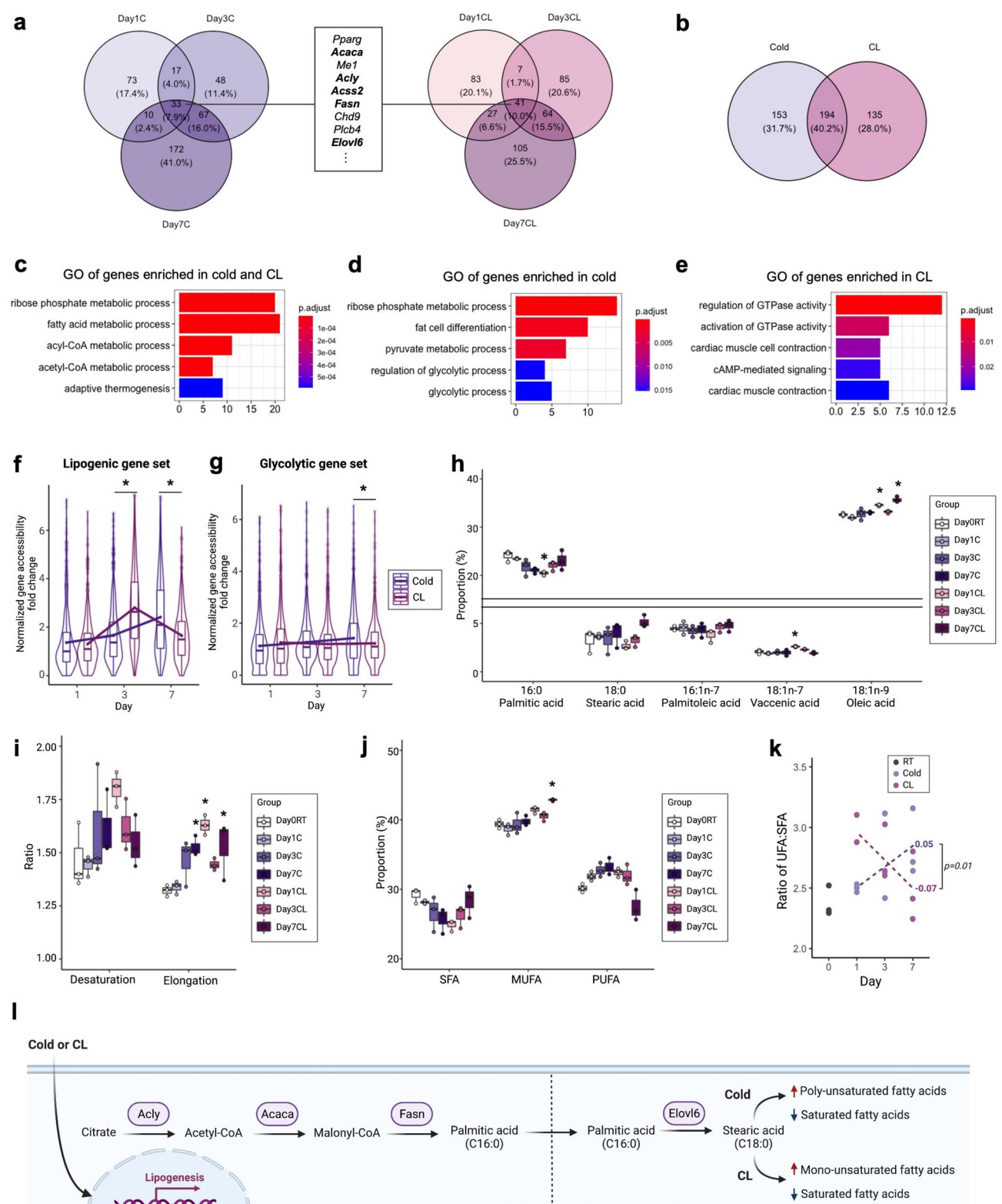

the same iWAT used for snATAC-seq by gas-chromatography mass spectrometry (GC-MS)-based lipid profiling. The most abundant free FAs were oleic acid (18:1n-9), linoleic acid (18:2n-6), and palmitic acid (16:0), making up around 82% of the FA content of the depot (Fig. 7h and Supplementary Fig. 8c). Palmitoleic acid (16:1n-7), stearic acid (18:0), and vaccenic acid (18:1n-7) were the next most abundant free FAs, which account for 10% (Fig. 7h and Supplementary Fig. 8c).

Based on the composition of major FAs and its elongation products, we calculated the metabolic activity of elongation and found that both cold and CL significantly increased the ratio of elongation (Fig. 7i). Consistently, our snATAC-seq analysis showed changes in FA elongation and accessibility of *Elovl6*, a gene encoding a key enzyme that catalyzes the elongation of FAs (Figs. 2h and 7a), providing evidence for the correlation between chromatin accessibility and their functional modulation of lipid

**Fig. 7 Shifts in lipogenic and glycolytic genes and lipid composition in response to different thermogenic stimuli. a** Venn diagram summary showing more accessible genes at day 1, 3, and 7 after cold exposure or CL treatment compared to Day 0 RT. The genes that are commonly more accessible after cold exposure and CL treatment from day 1 to day 7 are shown in the box. Bold font indicates the genes involved in *de novo* lipogenesis (*DNL*). **b** Venn diagram summary showing commonly or differentially more accessible genes after 3 or 7 days of cold exposure and CL treatment compared to Day 0 RT. **c** GO analysis of genes that were commonly more accessible in both cold and CL. **d, e** GO analysis of genes that are more accessible only in cold **d** or CL **e**. X-axis indicates number of genes. Color indicates adjusted *p*-value. **f, g** Violin and box plots showing the distribution of average gene accessibility of lipogenic genes (*Acaca, Acacb, Acly, Acss2, Fasn, Scd1, Elovl6*) **f** and glycolytic genes (*Hk1, Gpi1, Pfkl, Pfkp, Gapdh, Pgk1, Pgm1, Eno1, Pkm, Ldha*) (**g**) of fold changes after cold exposure and CL treatment compared to Day0 RT. The lower and upper hinges indicate the 25th and 75th percentiles. The horizontal line in the middle of box plot denotes the median. The mean for each day for cold and CL are connected by a line. *\*p*-value<0.05, Welch's two sample t-test was performed. **h** Relative concentration of quantified lipid species after cold exposure and CL treatment (*n* = 3 per group; More lipid species in Supplementary Fig. 8c). **i** Ratio of desaturation and elongation from lipidomics data (*n* = 3 per group). **j**, Relative proportion of saturated FA (SFA), monounsaturated FA (MUFA), and polyunsaturated FA (PUFA) by lipidomics analysis (*n* = 3 per group). **k** Ratio of unsaturated FA (UFA) to SFA (*n* = 3 per group). The slope for each cold and CL from day 1 to day 7 is calculated by a linear regression function. The *p*-value for the comparison of the slopes between cold and CL is indicated. **l**, Model for the contribution of cold temperature and CL treatment to FA metabolic process through functional accessibility changes at lipidomic genes and changes in lipid profile. *\*Adjusted *p*-value<0.05, ANOVA multiple comparisons test with Bonferroni's post-hoc test was performed for **h**–**j**.

profiles. Next, we characterized changes in lipid composition after cold and CL treatment. Despite the increased elongation by both cold and CL, different types of FAs were increased by cold and CL. Strikingly, CL treatment significantly increased the proportion of oleic acid, a major component of monounsaturated FA (MUFA), not cold treatment (Fig. 7h, j). In addition, the proportion of some lipid species have opposite dynamics between cold and CL. For example, cold exposure tended to decrease palmitic acid gradually over time course, while CL treatment dramatically decreased palmitic acid at day 1 then restored to the day 0 level by day 7 of CL treatment (Fig. 7h). Accordingly, the ratio of unsaturated FA (UFA) to saturated FA (SFA) had opposite trends between cold and CL over the time course of our study (Fig. 7k). Overall, these results reveal that both cold and CL treatment increase abundance of longer chain FAs in adipose, but they increase different types of UFA in adipose with the different dynamics (Fig. 7l).

## Discussion

Here we provide first comprehensive atlas of iWAT at chromatin status at single-nucleus resolution in response to two major thermogenic stimuli. Our analyses identified changes in cell abundances, patterns of differential gene accessibility across the time course of the treatments, beige-specific enhancers and TF motif modules regulating beige adipocyte development, and a pseudotime trajectory for beige adipocyte development. These changes created functional consequences in FA metabolism detectable by lipidomics analysis. Collectively, our data report the genes and GRNs underlying the developmental emergence of beige adipocytes, as well as the metabolic consequences in adipose during responses to cold exposure and CL treatment.

Mature adipocytes underwent the most profound changes, with a beige adipocyte population accumulating in iWAT with increased exposure time of both stimuli (Figs. 1d–f and 2d). The abundance of beige adipocytes increased to a similar level between cold and CL (Fig. 1d), however, the abundance of total adipocytes was in the range of 18% to 23% after cold exposure and about 20% to 37% after CL treatment (Supplementary Fig. 4b). This may indicate that CL generates more white adipocytes which can be further *trans*-differentiated into beige adipocytes. This supports the notion that Adrb3 activation by CL initiates beige adipocyte formation by converting pre-existing white adipocytes into beige adipocytes[6].

In addition to detecting beige adipocyte emergence during cold and CL responses, we detected two major APCs in iWAT. Both APCs had higher accessibility at *Pdgfra* (Fig. 5b). APC1 had higher accessibility of genes for stem cell surface markers such as *Ly6a*

(Sca1) and *Cd34*; whereas, APC2 had higher accessibility at mural cell marker genes including *Pdgfrb* and *Acta2* (Fig. 5b). Our results are consistent with a recent lineage tracing study identifying two major adipocyte progenitors, *Pdgfra*+ and *Pdgfra*+/*Pdgfrb*+, using dual recombinase-mediated genetic labeling[61]. The genes upregulated in *Pdgfra*+/*Pdgfrb*+ cells were associated with wounding responses and likely correspond to APC2 in our snATAC-seq data (Fig. 5c)[64]. We originally hypothesized that the origins of adipocytes would be revealed by characterizing intermediate cells between clusters, which may represent transition states of cells differentiating from their precursors. However, linkages representing transition cells were not observed in the UMAP embedding from our snATAC-seq data. We reasoned that this could be due to the small number of APCs in our data. Therefore, our inference of adipocyte trajectory was limited to mature adipocyte populations. Further investigation into transition states of cells from progenitors and genetic memory could help elucidate how new adipocytes are generated in response to thermogenic stimuli.

Several lines of evidence suggest that beige adipocyte progenitors reside among mural cells within the vasculature[10,65]. We found that stromal cell abundance tended to increase with cold and CL treatment (Fig. 1d). Our results show that the vasculature expansion is influenced by VEGF signaling between adipocytes and VECs through the *Vegfra* and *Vegfr1* ligand-receptor pair (Fig. 5e). An important feature of this mechanism may involve PGC-1α and ESRRα binding motifs, which were enriched in beige adipocytes, and were shown to be important in skeletal muscle angiogenesis by *Vegfa*[66].

Besides detecting adipocytes, their precursors and vascular cells, we detected dendritic cells and macrophages in iWAT, and their reductions during the exposure intervals (Fig. 1d). Evidence indicates these immune cell reductions are important to the adipocyte changes we report. First, NF-kB pathway activation in dendritic cells stimulates proinflammatory gene expression and inflammatory responses leading to metabolic disorders in insulin sensitive tissues[61,67]; accordingly, reduced inflammatory cell abundances after CL and cold exposure may limit the appearances of these disorders. Second, knockout of IκB kinase ε (IKKε) which is upregulated by NF-kB was shown to promote thermogenesis and prevent HFD induced obesity in mice[68]; accordingly, fewer cells utilizing NF-kB signaling after CL and cold exposure may also promote thermogenesis. Collectively, these results suggest that both cold and CL promote thermogenesis and prevent activation of NF-kB pathway in adipose tissue by reducing dendritic cells in adipose tissue.

An important question in thermogenic stimuli induced beiging is how changes in adipose tissue differ in response to two most

widely used thermogenic agents, cold and CL treatment. Pseudotime trajectories were quite similar for developing beige adipocytes emerging upon both CL and cold exposure (Supplementary Fig. 6a, b). Additionally, the fold-changes in gene accessibility after cold and CL treatment compared to RT were positively correlated for the two treatments (Supplementary Fig. 9a, b). Despite these similarities, there were notable differences in responses to the two thermogenic stimuli. Genes involved in glycolysis were relatively more activated by cold stress (Fig. 7d, g). Specifically, Eno1 was highly accessible in cold-induced beige adipocytes (Supplementary Fig. 9c). This is consistent with the findings from Chen et al. who reported that many glycolytic genes were up-regulated in a subset of beige adipocytes formed during cold adaptation in the absence of β-AR signaling[56]. In contrast, CL treatment activated gene programs associated with cAMP responses and regulation of G-proteins relatively more compared to cold (Fig. 7e and Supplementary Fig. 9d–i). These findings suggest that cold and CL exert their effects through distinct mechanisms, with cold triggering extra beige fat formation via β-AR independent changes in regulation of genes involved in glucose metabolism.

A key finding from our analyses was that the genes involved in DNL exhibited markedly increased accessibility after both cold and CL treatment for 1, 3, and 7 days (Fig. 7a and Supplementary Fig. 8a). In agreement with our results, 4 days of cold exposure and CL treatment also increased gene expression levels of lipogenic genes in snRNA-seq data (Supplementary Fig. 8b)[9]. These results suggest that induction of beiging is accompanied by an activation of lipogenesis. Lipogenesis may contribute to thermogenesis through futile cycling of FA synthesis and oxidation by maintaining the availability of endogenous FAs as fuel for heat production[69,70]. Indeed, the finding that Pdk4 was enriched in beige adipocytes may be relevant to shifting energy source from glucose to FA in beige adipocytes, resulting in an increased demand for lipids[45]. It is noteworthy that both FA synthesis (anabolic) and FA oxidation (catabolic) processes were activated after thermogenic stimuli, indicating these are not mutually exclusive responses.

Changes in chromatin accessibility at genes involved in FA metabolism led to functional alterations in lipid composition in adipose tissue. Not all FAs are oxidized at the same rates, which may explain the changes in the cellular accumulation of FAs. SFAs with longer chain length are oxidized at slower rates, while oleic acid among MUFAs, and α-linolenic acid (18:3n-3) and linoleic acid among PUFAs, are oxidized at faster rates[71]. We reported that CL significantly increased the relative abundance of oleic acid, whereas cold increased the relative abundance of linoleic acid. These results suggest that cold and CL change lipid composition of adipose tissue which have an influence on FA oxidation.

Our lipid analysis revealed that cold treatment tended to increase the relative abundance of PUFAs, and increased the UFA:SFA ratio from day 1 to day 7 (Fig. 7j, k). This may represent an important adaptive change in animals living in cold conditions. UFAs, and especially PUFAs, have lower melting temperatures relative to SFAs. The increased PUFA level seen in adipose in cold-exposed mice may contribute to preserving fluidity of storage lipids, enabling them to be readily accessed by lipases and used as energy sources[72]. Additionally, membrane phospholipids have higher fluidity when they are composed of UFAs, which may be important in cold temperatures[73]. These effects of cold exposure on lipid composition of adipose may be most important to poikilothermic animals, whose body temperatures varies with the environmental temperature[73], but they may be important for cutaneous and subcutaneous tissues in homeotherms, which maintain thermal homeostasis at internal tissues.

In addition to the differences we found between CL and cold that affect signaling and glycolytic pathways, there were notable chromatin state and lipid differences, some of which may underlie cardiac pathologies associated with CL treatments, contraindicating the use of this drug in people. First, consistent with reports that activation of Adrb3 affects cardiac contractility in human[12], we found that CL treatment opened chromatin at genes associated with cardiac muscle contraction (Fig. 7e). Interestingly, a recent study suggested that overstimulation by isoproterenol, a β1- and β2-AR agonist, promoted the release of palmitic acid, palmitoleic acid, and oleic acid from adipose tissue, which caused myocardial fibrosis and apoptosis[74]. We also reported an increase in the proportion of oleic acid after CL treatment, but not cold treatment (Fig. 7h). Collectively, our data might support the notion for an adipose tissue-heart communication in the development of cardiac diseases by regulating gene accessibility and lipid content in adipose tissue. Second, there were different kinetics in timing of gene accessibility changes and lipid species between cold and CL (Fig. 7f, g). Lastly, it was shown that treatment with oleic acid increased intracellular levels of cAMP in skeletal muscle cells[75]. Thus, it might be conceivable that oleic acid increase after CL treatment promoted the changes in accessibility of genes involved in cAMP signaling pathway. However, as CL activates Adrb3 receptor, a major pathway activating thermogenesis in beige cells, it is possible that these differences were observed due to different effects/dosages of endogenous norepinephrine induced by CL. Additionally, there may be differences in kinetics of the responses to CL and cold exposures that are read out as chromatin state differences. Future studies will be needed to demonstrate the effects of lipids and metabolic properties of adipocytes changed by cold and CL treatment.

Our analyses were mainly based on chromatin accessibility which may or may not be directly linked to the transcription state. Future studies would benefit from recent techniques such as SNARE-seq[76] for joint profiling of chromatin accessibility and gene expression in the same cells or nuclei. Nonetheless, publicly available snRNA-seq data were used to pinpoint the specific changes in our analyses. Transcriptional changes may or may not lead to changes in translated protein, or protein activities. By assaying lipid profiles in the same tissues used for chromatin analysis, we were able to characterize functional physiologic and metabolic changes that accompanied gene accessibility changes. In summary, this study highlights underlying gene regulatory mechanisms for beige adipocyte development and the common and distinct properties of cold and Adrb3-induced beige adipocytes. These findings could provide pivotal insight into clinical utility of environmental and pharmacological beige stimuli.

## Methods

**Animals**. All animal experiment was in accordance with approved protocols by the Institutional Animal Care and Use Committee (IACUC) at Cornell University (Protocol number 2017-0063). Six-weeks-old C57BL/6 J mice were purchased from Jackson Laboratories. Mice were acclimated to the facility and maintained at room temperature (RT, 22 °C) for two-weeks before intervention. Mice were randomly assigned to cold (6 °C) or CL-316,243 (CL, 1 mg/kg), a β3-adrenergic receptor (Adrb3) selective agonist treatment group. Mice were exposed to cold and Adrb3 agonist for 0, 1, 3, or 7 days.

**Histological staining**. Adipose tissues were fixed in 10% formalin, paraffin embedded, and sectioned at 5μm thickness with a microtome. Slides were deparaffinized, rehydrated, and stained with hematoxylin and eosin (H&E).

**Tn5 transposase purification & loading**. Tn5 was produced as described with no modifications[77]. Tn5 transposase was generated by assembling pre-annealed ME-A and ME-B (IDT, standard desalting; Supplementary Table 1). Oligonucleotides (IDT, standard desalting) were annealed at 95 °C for 2 min followed by cooling to 25 °C at a 0.1 °C/s cooling rate. Annealed ME-A and ME-B oligos were each mixed

in a 1:1 ratio Tn5 transposase and incubated for 30 min at room temperature. Eight A-transposomes and twelve B-transposomes were formed to generate 96 unique transposome combinations.

**Nuclei isolation from adipose tissue**. Inguinal adipose tissues were collected from mice before any intervention and after 1, 3 or 7 days of cold exposure or CL treatment and frozen at −80 °C. All steps were performed on ice or at 4 °C. To avoid perturbations caused by isolating cells, whole adipose tissues were used to extract nuclei without isolating mature adipocytes or SVF. Whole adipose tissues were minced on dry ice. 80-100 mg of chopped adipose tissue were transferred to a chilled 40 mL Dounce Homogenize containing 25 mL homogenize buffer (320 mM sucrose, 0.1 mM EDTA, 0.1% NP40 (28324, ThermoFisher), 5 mM $CaCl_2$, 3 mM $Mg(Ac)_2$, 10 mM Tris pH 7.8, protease inhibitors (88666, Pierce), 0.016 mM PMSF, 0.33 mM β-mercaptoethanol)[78]. Tissue was homogenized immediately via 10 gentle strokes of the loose pestle and then 10 gentle strokes of the tight pestle on ice using a Dounce Tissue Grinder (357546, Wheaton). The homogenate was filtered through a 40 μm nylon mesh (10199-658, VWR) and centrifuged for 10 min at 500 g. The top fat layer was discarded, and the supernatant was transferred into a new tube without disrupting the pellet. The samples were then resuspended in 5 vol. of ice-cold ATAC-Resuspension Wash Buffer (10 mM Tris-HCl pH 7.4, 10 mM NaCl, 3 mM $MgCl_2$ and 0.1% Tween 20 in water) and centrifuged for 5 min at 500 g in a swinging bucket rotor centrifuge (5920 R, Eppendorf). Nuclei pellets were resuspended in 1X Tagmentation Buffer (10 mM Tris pH 7.4, 5 mM $MgCl_2$, 10% DMF, 33% 1X PBS (without Ca++ and Mg++), 0.1% Tween-20, 0.01% Digitonin (BN2006, ThermoFisher)). The number of nuclei was counted using a hemocytometer.

**snATAC-seq with combinatorial indexing**. Nuclei concentration was adjusted to 160,000 nuclei/mL in 1X Tagmentation Buffer. 8 μL of nuclei were distributed into each well of a 96-well plate (1,280 nuclei/well) containing 1 μL of each ME-A or ME-B carrying barcoded transposome per well at ~1.5 μM. Tagmentation was performed at 50 °C for 30 min. Following tagmentation, 10 μL of 40 μM EDTA was added each well and the plate was incubated at 37 °C for 15 min to terminate the Tn5 reaction. Next, 20 μL of sort buffer (2% BSA and 2 mM EDTA in PBS) was added to each well. All wells were pooled and centrifuged for 5 min at 500 g. Nuclei were resuspended in sort buffer with 10% DMSO and frozen at −80 °C until ready for fluorescence activated nuclei sorting (FANS).

After thawing at 37 °C for 1 min, nuclei were centrifuged for 5 min at 500 g and resuspended in 1 mL of sort buffer. Nuclei were filtered through a 35 μm mesh (352235, Corning) and stained with 3 μM Draq7 (ab109202, Abcam) before FANS. 25 Draq7+ nuclei were sorted into each well of second 96-well plates containing 16.5 μL of elution buffer (2% BSA and 10 mM Tris pH 8.0) using a BD FASC Melody. Gating was selected to isolate single Draq7+ nuclei. After sorting, plates were frozen at −80 °C.

For PCR amplification, 2 μL of 0.2% SDS were added to each well and the plate was incubated for 7 min at 55 °C to denature transposase. 2.5 μL of 10% Triton X-100 were added to each well to quench SDS. 1.5 μL of 25 μM Primer i5 and 1.5 μL of 25 μM Primer i7 (Supplementary Table 1) were added to each well and the plate was gently vortexed and spun down briefly. 25 μL of PCR mix (Q5 DNA polymerase (M0491, NEB), 2 mM dNTP, Q5 buffer and 1X GC Enhancer) were added to each well. PCR was performed using the following protocol: 72 °C 5 min, 98 °C 30 s, 13 cycles of: [98 °C 10 s, 63 °C 30 s, 72 °C 30 s], 72 °C 5 min, held at 4 °C. All wells were pooled. Amplified DNA libraries were purified using MinElute columns (28004, Qiagen) using a vacuum apparatus (19413, Qiagen), washed twice with 750 μL Buffer PE (19065, Qiagen) and spun down at maximum speed for 1 min. Samples were eluted twice with warm elution buffer (10 mM Tris pH 8). The size selection was performed using Ampure XP Bead (A63880, Beckman Coulter). The concentration of each pooled PCR plate library was measured using a Qubit dsDNA HS Assay Kit (Q32851, Invitrogen) and library fragment distribution was quantified using an Agilent Bioanalyzer. Libraries were subjected to digital PCR on a Bio-Rad QX200 droplet digital PCR. Libraries were loaded at 8 pM on a mid-lane output PE 150 bp. Libraries were sequenced on a Nextseq500 sequencer (Illumina) using custom read primers (Supplementary Table 1); custom recipe for the following, read 1: [36 imaged cycles], Index 1: [8 imaged cycles, 27 dark cycles, 8 imaged cycles], Index 2: [8 imaged cycles, 21 dark cycles, 8 imaged cycles], Read 2: [36 imaged cycles].

**snATAC-seq data pre-processing**. Fastq files from two sequencing runs were merged and demultiplexed by adding cell barcode to each read. Demultiplexed pair-end reads were aligned to mm10 using Bowtie2[79] with the parameters: bowtie2 -p 16 -t -X 2000-no-mixed-no-discordant. After alignment, reads were sorted by read name using samtools. The pair-end reads with low mapping quality fragments < MAPQ 30 and improperly paired fragments (SAM flag = 1804) were removed. Reads were separated based on the cell barcode and reads belonging to the same cell barcode were deduplicated. Of 1,020,363,364 sequenced read pairs, 791,158,286 (77.5%) mapped to the reference genome, with an assigned cell barcode. PCR duplicate reads (35%) and mitochondrial reads (0.2%) were removed. To check the quality of library, frequency of the insert sizes representing nucleosome patterns was plotted using ATACseqQC[80] from aggregated samples (Supplementary Fig. 2a).

**snATAC-seq data quality control**. To select cell barcodes, a histogram of the log-transformed number of reads per cell barcode was plotted (Supplementary Fig. 2d). The bimodal distribution indicates that cell barcodes with a low read sequencing depth are improper barcodes with background reads and cell barcodes within a high read depth distribution of reads are genuine cells. For initial filtering, we filtered out cells which have less than 1,000 reads (Supplementary Fig. 2c). To minimize the potential bias caused by the most abundant cell populations using aggregated accessibility peak sites from all cells, we adapted the previous strategy to generate a merged peak set from diverse cell populations after initial clustering[13]. For initial clustering, we first generated a cell-by-10kb-window sparse matrix using snapATAC[81]. To create this matrix, the genome was broken into uniformly sized 10 kb windows and the number of reads at a given bin for each cell was counted and binarized. Any windows overlapping with the ENCODE mm10 blacklist[82] were filtered. Then, we used Signac, an R toolkit extension of Seurat for the analysis of single-cell chromatin data, for further analysis[21,22].

Next, we normalized the cell-by-windows matrix by a TF-IDF transformation. Then, we performed shared neighbor network (SNN) graph-based clustering with the top 25 principal components except for 1st principal component. The 1st component was excluded for dimensionality reduction because it has a high correlation with sequencing depth. We visualized cell clusters by Uniform Manifold Approximation and Projection (UMAP). After identifying initial clusters, all cells in individual clusters were aggregated. For each cluster, peak calling was performed using the MACS2[83] 'callpeak' command with the parameters: -nomodel-shift −100-extsize 200-keep-dup-all -q0.05. The summits were extended ±250 bp and the peak sites overlapping with the ENCODE mm10 blacklist[82] were filtered. The peak sites identified from each cluster were merged and used to create a cell-by-peak matrix. Cells with less than 0.15 reads in peak ratio were removed. Then, we repeated clustering based on the cell-by-peak matrix. To create a gene activity matrix, we counted the reads on gene coordinates for mouse genome from EnsembleDB extended to include the 2 kb upstream[84].

**Identification of doublets**. Initially, we did not exclude doublets to see an impact of the doublets in our analysis. The major cluster identification was not significantly affected (Supplementary Fig. 2e). However, when we subclustered a single cluster, we consistently observed that a small fraction of cells formed a distinct cluster (Supplementary Fig. 2g, r). As an example, we observed a small cluster (cluster 2) among adipocytes (Supplementary Fig. 2g). GO analysis showed that the genes relatively more enriched in cluster 2 are associated with lymphocyte proliferation (Supplementary Fig. 2i). While accessibility of *Adipoq* was similar between cluster 1 and cluster 2, accessibility of *Ptprc* was relatively higher in cluster 2; the cluster 2 having intermediate accessibilities of marker genes for adipocytes and lymphocytes (Supplementary Fig. 2j, k). However, adipocytes and lymphocytes are less likely to share similar features. For example, PPARA motif enriched in adipocytes and ETS1 motif enriched in lymphocytes showed a negative correlation (Supplementary Fig. 2l). Therefore, we further analyzed if the cluster 2 exhibiting a hybrid feature of adipocytes and lymphocytes, the two largest clusters in our dataset, is a putative collision cluster.

The putative collision cluster could appear due to cell barcode collisions since the combinatorial indexing strategy procedure may generate barcode collision as previously reported[13]. Thus, it is likely to misinterpret a putative collision cluster as a novel cluster if doublets are included for further analysis. To determine whether the putative collision cluster is a novel cluster or a collision cluster, we first calculated the number of reads per cell. However, unlike the assumption that barcode collision cells have higher library complexity as they have reads from two different cells, the putative collision cluster showed the average library complexity as other subclusters in our dataset (Supplementary Fig. 2m, n). We generated 'in-silico collisions' by combining reads from random pairs of cells from our dataset. We found that these in-silico collisions clustered together with putative collision clusters and take similar coordinates on UMAP. To measure a likelihood of a cell being a doublet/collision, we adapted Scrublet[85], a scRNA-seq tool for identifying doublets by calculating fraction of a cell's neighbors that are in-silico doublets. The cells clustered into a putative collision cluster consistently showed high doublet scores, indicating these cells have high likelihood of being collision cells (Supplementary Fig. 2o, p). We also confirmed that the cells found in the putative collision cluster, doublets identified by Scrublet, and in-silico collisions have high correlations compared to singlets (Supplementary Fig. 2q). Overall, we filtered out 2,956 cells (8.3%) with a doublet score higher than 0.2 cutoff. It is consistent with the collision rate (~9%) estimated from our previous mixed species experiment[86]. After removing doublets, the putative collision clusters were no longer detected. Therefore, we performed further analysis without doublets.

**Identification of cell clusters and cell type annotation**. A total 32,552 cells were used for further analysis after filtering out the low-quality cells (reads per cell < 1,000, reads in peak ratio < 0.15, and doublet score > 0.2). We repeated nonlinear dimension reduction and cell clustering based on top 50% variable peaks and 2nd to 15th components using Seurat's SNN graph clustering[87]. To resolve identities of the cell clusters, we first identified sets of genes or peaks enriched in individual cell clusters in comparison with all other cell clusters by differential accessibility analysis using 'FindAllMarker' or 'FindMarker' function in Signac[21]. For the gene-based analysis, gene activity quantified by summing the fragments intersecting gene

body and 2 kb upstream region was used. GO and KEGG pathway analyses were performed on a set of enriched genes (adjusted p-value or FDR < 0.05 and |log2FC| > 1.2) with ClusterProfiler[88]. In addition, Genomic Regions Enrichment of Annotations Tool (GREAT)[25] analysis were performed on differentially accessible peaks.

After annotating cell types based on snATAC-seq data, we confirmed the cell type annotation with publicly available scRNA-seq data of SVF and snRNA-seq data of mature adipocytes from iWAT[9]. To transfer cell labels from the reference scRNA-seq and snRNA-seq data, scRNA-seq and snRNA-seq datasets were integrated with our snATAC-seq using SCTransform after normalizing two datasets[87]. For general annotation, we merged subpopulations of the same cell types in scRNA-seq data to one major cell type (e.g., adipocyte 1 through adipocyte 16 into one adipocyte cluster). Almost 85% of cells received higher than 0.5 prediction confidence scores except for the clusters such as vascular smooth muscle cells which were not found in scRNA-seq data.

**Cell abundance test with Milo**. In addition to evaluating cell proportion changes among discrete clusters, we performed Milo[24] to test differential abundance between experimental conditions (RT at day 0 and after cold exposure or CL treatment). Milo identified neighborhoods based on a k-nearest neighbor (KNN) graph and assigned cells to the neighborhoods[24]. Then, the number of cells in neighborhoods were counted and a matrix of the number of neighborhoods by the number of experimental samples was created. This revealed differential abundance between treatment conditions in a cell state rather than a pre-defined cell cluster.

**Transcription factor motif analysis with chromVAR**. Firstly, to analyze differentially active motifs, motif analysis was performed on a set of peaks enriched in one group of cells compared to the other groups of cells using a function implemented in Signac[21]. Next, TF motif activity was computed by chromVAR and a cell-by TF z-score matrix was created[46]. To identify transcriptional programs underlying the cell states during beiging among 726 TFs in JASPAR2020 database[40], we measured Pearson correlation between motifs using motif activity score computed by chromVAR. 64 TF motifs with motif variance higher than 1.8 were used for unsupervised hierarchical clustering based on the correlation coefficients to identify modules.

**Co-accessibility with cicero**. To predict the potential *cis*-regulatory interactions, cicero was used to construct *cis*-regulatory networks by assessing co-accessibility of pairs of DNA elements and linking a promoter and its potential enhancers in distal sites[26]. Cicero calculated correlation (co-accessibility) between putative regulatory elements from aggregated accessibility across groups of cells. Then, links between regulatory elements and target genes were identified as putative enhancer-promoter pairs.

**Inference of cell-cell interaction with CellChat**. To predict the potential cell-cell interaction from our snATAC-seq data, we utilized gene accessibility of ligands and receptors from a literature-supported signaling molecule interaction database ('CellChatDB')[58]. Then we predicted significant communications by identifying differentially accessible ligands and receptors for each cell cluster and associating each interaction with a probability value[58]. Then the significant cell-cell communications were identified by a statistical test that randomly permutating cell cluster and then recalculate the interaction probability[58].

**Pseudotime trajectory analysis**. We used ArchR[89] to calculate the pseudotime value for each cell and to create pseudotime trajectory in a two-dimensional space. The adipocytes that passed previous quality control in Signac were used without additional filtering in ArchR. The trajectory backbone was defined to provide a rough ordering of cells from the cluster including the most day 0 cells to the other cluster with the most day 7 cells. Next, ArchR calculated mean coordinates of each cluster and computed the distance to the mean coordinates for each cell. ArchR provided a pseudotime value for each cell based on the Euclidean distance to their mean coordinates and plotted a trajectory. Finally, ArchR analyzed changes in features across pseudotime in motifs and genes.

**Lipidomics**. A total of 40–50 mg of chopped adipose tissue was subjected to lipid extraction using a modified one step lipid extraction method to obtain fatty acid methyl esters (FAME)[90]. Adipose tissue was heated with an aqueous digesting and methylating reagent (1.2 mL methanol, 0.15 mL 2,2-dimethoxypropane (DMP) and 0.05 mL H2SO4) and an organic extraction reagent (1 mL heptane, 0.6 mL toluene) for 2 h at 80 °C. After cooling the tube at room temperature, 2 mL heptane and 2 mL saturated NaCl were added to the sample and centrifuged for 10 min at 3500 rpm. After keeping the sample at 4 °C overnight, two phases were formed, and top layer was transferred to a new tube. The sample was dried down under nitrogen and 60 μL of heptane was added. FAME were quantified by Shimadzu GCMS-TQ8050 triple quadrupole mass spectrometer with a CI-MS (Shimadzu) with a BPX 70 column (25 m × 0.22 mm × 0.25 m; SGE Inc.). An equal weight FAME mixture (462 A; Nu-Chek Prep, Inc.) was used to calculate response factors. The FAME were analyzed by gas chromatography covalent adduct chemical ionization tandem mass spectrometry by molecular weight and by diagnostic ion analysis. Elongation ratio was estimated by calculating $(18:0 + 18:1n\text{-}7 + 18:1n\text{-}9)/(16:0 + 16:1n\text{-}7)$. Desaturation ratio was estimated by calculating $(18:1n\text{-}7 + 18:1n\text{-}9 + 16:1n\text{-}7)/(16:0 + 18:0)$.

**Statistics and reproducibility**. All statistical tests in our analysis were performed using R version (v3.5.0) and associated packages. In our analysis, cell proportions in Fig. 1d and lipid proportions in Fig. 7h–j were compared by ANOVA multiple comparisons test with Bonferroni's post-hoc test. In the case of comparing two treatment conditions in Fig. 7f, g, Welch's two sample t-test was performed. Comparisons were two-sided unless otherwise noted. Alpha was set at 0.05.

**Reporting summary**. Further information on research design is available in the Nature Research Reporting Summary linked to this article.

## Data availability

snATAC-seq data reported in this study is deposited in NCBI Gene Expression Omnibus under accession number GSE185377. Publicly available snRNA-seq data of adipocytes and scRNA-seq data of adipose SVF from mice treated with saline, cold or CL up to 4 days were obtained from GEO with the accession number GSE133486[9]. The GEO accession number for PGC-1α ChIP-seq data from mouse brown adipose tissue is GSE110056[39]. The publicly available ESRRα ChIP-seq data from mouse liver was obtained from GEO with the accession number GSE43638[42]. The MEF2α ChIP-exo data from mouse skeletal myoblasts were obtained from GEO with the accession number GSE61207[43]. H3K27ac and H3K4me1 ChIP-seq data from beige adipocytes after cold exposure and white adipocytes at 30 °C were obtained from GEO with the accession number GSE108077[27]. All other data are available from the authors upon reasonable request.

## Code availability

Code utilized in the study is immediately available from the authors upon reasonable request.

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

## Acknowledgements

This work was supported by National Institutes of Health (NIH) grants K01 DK109027. We thank the Cornell University Biotechnology Resource Center (BRC) for their assistance with FACS sorting and sequencing. We thank Paul Cohen at the Rockefeller University and Bethany Cummings at the University of California Davis for their discussions, and Debadrita Bhattacharya at the Cornell University for her help on the TF module analysis.

## Author contributions

S.L., P.D.S., and D.C.B. conceived this study. S.L., P.D.S., D.C.B., and J.T.B. designed the experiments. S.L. carried out the snATAC-seq library preparation and data analyses. A.B. carried out the mouse experiment. H.G.P. performed the lipidomics experiment. R.S. and B.H. contributed to snATAC-seq sample preparation and provided suggestions on the data analysis. S.L. wrote the draft and S.L., P.D.S., and D.C.B. edited the manuscript. P.D.S. supervised this study.

## Competing interests

The authors declare no competing interests.
