## [Peer Review File · Communications Biology]

Reviewers' comments:

Reviewer #1 (Remarks to the Author):

In this paper submitted for review to Communications Biology, the authors characterized and compared chromatin accessibility using snATAC-seq for white adipose tissue under cold- and CL-stimulated conditions. From this analysis, the authors were able to show differences in the regulation of lipogenic gene sets during extended cold exposure in CL316,243 treated mice relative to cold-treated mice resulting in changed lipid composition. The dataset generated would be informative to the field. Additional issues discussed below could be addressed to improve the manuscript for readers.

- Figure 2 – The subpopulations need to be better defined. Are the clusters initially defined such that white adipocytes and beige adipocytes belong to the same cluster and the authors specifically forced a subclustering of an adipocyte cluster? Or were they initially present in the clustering algorithm as two independent clusters?

- Figure 4 – Was the pseudotime analysis restricted to just the adipocytes? Since the beige adipocytes can arise from both transdifferentiation of unilocular mature adipocytes and de novo differentiation from progenitors (lines 231-232), it would be appropriate to include APCs in the pseudotime analysis?

- Figure 4 – Since the manuscript focuses on comparing the effects of cold exposure vs. CL treatment, it would be important to determine whether the TF modules 1, 2 or 3 were differentially regulated by cold or CL over time?

- The method for how ligand-receptor pairs were determined was not adequately explained.

- The authors should be more consistent and more explicit when reporting statistical differences.

In many cases, the authors treat trends towards a difference as a statistical difference. For example, in line 390, the authors indicated, "The proportion of palmitic acid... was decreased after CL and cold treatment," while only CL treatment has a significant difference.

- The characterization of individual cell types was at a superficial level. For example, the major conclusions of Figures 5 and 6 are not clear except that vascular and endothelial cells expressing vascular and endothelial cell markers.

- A primary concern is that the follow-up studies described in Figure 7 are generally unconvincing. If the major difference between cold and CL316,243 treatment was lipogenic gene expression, the manuscript would benefit from qPCR or other validation. Also, these studies have a relatively low number ($n = 3$). Overall, it is difficult to interpret the relevance of these follow-up studies have on beige adipocyte physiology.

o Although it can be common for GC/MS analysis of FAs to be represented as percentages rather than absolute values, it is difficult to determine whether these changes are due to relative FA production changes or preferential β -oxidation of some FAs over others. It would help if the authors could include the weight of the adipose tissue next to the data. Additionally, the authors could present the data as a stacked bar graph where all the columns add up to 100%. This would highlight that the data is a percentage rather than an absolute.

o What significance does the different fatty acid compositions have on a functional level? The paper would be substantially stronger if the author could discuss the different fatty acid compositions and related adipocyte functions.

o The authors mention the role of palmitic acid in inflammation in their discussion. In general, the inflammatory response is observed with the free fatty acid version of palmitate, whereas the authors are using gas chromatography to investigate both free and esterified versions of palmitate. This should be either further clarified or removed.

Also, the presentation of the data and the analysis could be refined for ease of reading and interpretation (specific examples are given below).

- Inguinal Adipose Tissue in Figure 1a is missing an "I"

- In multiple cases (including Figure 1d), the figure description says individual samples are represented with dots – however, I cannot see those individual samples.

- The resolution on Figures 1 makes it difficult to discern the legends on Figures 1e and 1f (For example, I cannot tell what the difference between an overlap size of 25, 50, or 75 is).

Additionally, it is not clear what plotting the overlap size, or the neighborhood size adds to the interpretation of this graph. Generally, all plots seem to have some labels that appear a bit small

to read easily.

- I think it is a mistake, but Figure 3d appears primarily black.
- Figure 4 describes 3 modules for understanding beige adipocyte development with individual genes representing each module appearing in Figure 4c. These genes could be labeled with their respective modules for clarity. Also, the trajectory in panel c is misspelled
- Figure 6 has many misspellings in the title of the figure. Also, b has an overlapping title on the right.
- Lines 52-54 – This sentence appears to be missing a word or two and is generally a little confusing. I would recommend splitting it into two sentences for clarity.
- In the introduction, I think the authors assume that pharmacologic administration of ADRB3 agonists is more practical in clinical settings than cold exposure. If so, they should specifically state it.
- Line 100 – there is a 22 that appears randomly (I assume it is a miss-formatted reference?)
- Lines 110-113 – The authors discussed the top marker genes enriched in white adipocytes and beige adipocytes. They should specify whether these genes were used to determine the identity of these subclusters or whether there was a different method for determining their identity.
- Discussion of some of the figures (primarily Fig 2c) is out of the order that they appear.
- The authors could explain the differences between KEGG and GO to explain why they used these different techniques.
- Lines 372-373 – if this is an important point, then the authors should include the panel in the main figures and not the supplementary figures.

Reviewer #2 (Remarks to the Author):

This study explores the chromatin accessibility dynamics of iWAT during cold and ADRB3 agonist-induced beiging at single-cell level. Beige adipocytes are the brown-like fat cells that can be recruited in WAT depots by cold and ADRB3 agonists. Distinct cell populations and mechanisms have previously been reported involving in beiging that induced by the two stimuli. In this present study, the authors further explored the chromatin accessibility change of WAT mediating cold and ADRB3 agonist-induced beiging. Twelve cell populations were identified based on the sn-ATAC, and the white and beige subpopulations were further examined. This leads to elucidation of changes in cell populations and chromatin accessibility under these two stimuli, and associated changes in regulatory pathways and fatty acid composition. This study employs a new technique to address an interesting biological question.

1. While the reviewer appreciates the use of sn-ATAC seq to generate a comprehensive atlas of chromatin status at single cell resolution under thermogenic stimuli, and the concern is that the rationale for dissecting the distinct mechanisms underlying cold and ADRB3 induced beiging is not well established. The action of these stimuli includes recruitment of new beige adipocytes and activation of existing beige adipocytes. As these stimuli may recruit beige cells from distinct (progenitor) cell types, it is obvious that the chromatin accessibility may be quite different as you may be comparing apples to pears. From the standpoint of activation of existing beige cell, shouldn't cold and ADRB3-agonist similarly activate thermogenesis of beige cells via the ADRB3 receptor and downstream signaling? If so, any difference that was observed may be due to different effects/dosages of endogenous NE (induced by cold) and exogenous ADRB3 agonist. Some discussion should be included to address these issues. The title is also misleading as the one on the PDF file is different from that in the one manuscript form.
2. As chromatin accessibility may or may not be directly linked to gene transcription, the currently results can be compared to publicly available snRNA-seq results to pinpoint the specific changes. Discussion on the limitation of sn-ATAC and how the limitation can be overcome by other technique should be included. For example, SNARE-seq can reveal both transcriptome and chromatin accessibility within the same cells.
3. The Introduction stated that the object of this study was to explore the difference of chromatin accessibility dynamics in cold or ADRB3 agonist-induced beige adipocytes, but the beige-specific transcription factor activity and chromatin interaction networks (Fig. 3) were only analyzed

between white and beige, the effect of treatments at various timepoints were not considered. Similar comparisons (between treatments) were also missing for the beige adipocyte developmental trajectory (Fig. 4). The only available comparison between cold and ADRB3 agonist treatment was for lipogenic and glycolytic genes (Fig. 7). However, the authors compared the whole adipocyte population, instead of beige adipocytes. The conclusion would be strengthened by including these missing comparisons.

4. The relative efficiency of cold and ADRB3 treatment used in this study on beiging should be validated by at least morphology of iWAT and immunoblot analysis of browning marker proteins, such as UCP1.

5. Fig.1d: The proportion of beige adipocyte under both treatments did not show significant increase. Can the authors explain this result?

6. Fig. 1a: No vehicle injection control was set for CL-316,243 injection group.

Reviewer #3 (Remarks to the Author):

This manuscript characterizes comprehensive cellular dynamics in adipose tissues during browning/beiging induced by two different thermogenic stimuli, cold exposure and an ADRB3 agonist (CL), by using snATAC-seq technique. The authors found common responses in the cell types involved in vascularization and inflammation for tissue remodeling. On the other hand, they showed differential changes in some metabolic processes in adipocyte populations, including glycolysis and lipogenesis. In particular, CL- and cold-induced adipocytes showed differential accessibility of lipogenic genes, which contributes to the distinct lipid composition in adipose tissues. These snATAC-seq data are not only confirming what has been known about adipose browning/beiging but also revealing some novel aspects. This is the first report of the use of snATAC-seq with combinatorial indexing on adipose tissue, which has a novelty value in methodology as well. Overall, it appears that the experiments were carefully performed, and the data were thoroughly analyzed. The data were mostly descriptive without functional validation tests, but this manuscript has significance in adipose biology and technical innovation.

There are a few minor issues that need to be addressed:

1. Despite using ATAC-seq chromatin accessibility data, most of the results are presented as genes, not as peaks. It is not clear how the peaks were assigned to genes. As this manuscript is missing with corresponding RNA-seq data, it is not shown and unknown how their snATAC-seq data would be actually reflected in gene expression. The authors should state this limitation and also clarify how the peak-to-gene assignment was performed. 1. Some statements about potential side effects of CL treatment in the Discussion are somewhat overinterpreting the results. For example, there is lack of data to link the increased MUFA content in adipose tissue to an increase in circulating MUFA.

2. Some of the Extended Data are not discussed in the text (e.g. Ext Data Fig. 4d-f). All the included data should be discussed or otherwise removed.

3. In Fig 4b & c, the label for pseudotime is confusing. If pseudotime starts from 0 to 100, it is interpreted as white adipocytes (colored blue) appearing from beige adipocytes (colored yellow), which is contrary to the color key in Fig 4b.

4. There are some typos and minor errors:

- Line 14: though -> through
 - Line 212: pseudotime -> pseudotime
 - Fig 3d: black background
 - p33, Fig 1d, there is no dot, while the legend states "Dots represent each biological replicate".
- Fig 1a: nguinal -> inguinal
- p34, Fig 2d labeling

COMMSBIO-21-3089 Response to Reviews

Reviewer #1:

Major comments:

1. Figure 2 – The subpopulations need to be better defined. Are the clusters initially defined such that white adipocytes and beige adipocytes belong to the same cluster and the authors specifically forced a subclustering of an adipocyte cluster? Or were they initially present in the clustering algorithm as two independent clusters?

They were initially identified as two independent clusters. Therefore, we reported 12 clusters in the Results (lines 93-94). We did not need to force the clustering algorithm to identify two separate adipocyte clusters.

2. Figure 4 – Was the pseudotime analysis restricted to just the adipocytes? Since the beige adipocytes can arise from both transdifferentiation of unilocular mature adipocytes and de novo differentiation from progenitors (lines 231-232), it would be appropriate to include APCs in the pseudotime analysis?

We agree with the reviewer's approach for the pseudotime analysis, and it was indeed what we originally did, along with including APCs in the pseudotime analysis. However, with our current data, there are a few limitations to forcing the APCs into the pseudotime trajectory as we described in the manuscript (lines 343-350). Firstly, there are two APC populations, but there is a lack of transition cells connecting them to mature adipocytes. This limits our ability to predict which cells are derived from which APC populations. Secondly, these APCs have the distant UMAP coordinates with very distinct feature profiles from mature adipocytes. when we attempted to include APCs in our pseudotime analysis, it captured very distinct features of adipocytes vs APCs, rather than the changes during early, intermediate, and late adipocyte development. Lastly, APCs were present in small numbers in our samples, which limits their impact on the trajectories. For these reasons, we restricted our pseudotime analysis to mature adipocyte populations only.

3. Figure 4 – Since the manuscript focuses on comparing the effects of cold exposure vs. CL treatment, it would be important to determine whether the TF modules 1, 2 or 3 were differentially regulated by cold or CL over time?

We thank the reviewer for the suggestion. Differentially accessible TF motif analysis between white and beige adipocytes for cold and CL individually showed the similar motif sets between cold and CL (Extended Data 5c,d). In both stimuli, the module 3 TF motifs were more enriched in beige adipocytes and the module 1 motifs were more enriched in white adipocytes in both stimuli. It was consistent with the TF module analysis. Modules 1,2, and 3 are preserved in both cold and CL. And we could not find significant differences between cold and CL in TF motif enrichment over pseudotime. Therefore, we used all adipocytes from both cold and CL to increase the statistical power.

4. The method for how ligand-receptor pairs were determined was not adequately explained.

We remedied this oversight by adding a paragraph to explain it in the methods section (lines 863-869).

5. The authors should be more consistent and more explicit when reporting statistical differences. In many cases, the authors treat trends towards a difference as a statistical difference. For example, in line 390, the authors indicated, “The proportion of palmitic acid... was decreased after CL and cold treatment,” while only CL treatment has a significant difference.

We agree with the importance of consistent and explicit wording. We have removed the misleading phrasing and added “tended” to the sentences where trends existed that were not statistically significant, and emphasizing where findings were statistically significant.

6. The characterization of individual cell types was at a superficial level. For example, the major conclusions of Figures 5 and 6 are not clear except that vascular and endothelial cells expressing vascular and endothelial cell markers.

We thank the reviewer for noting this concern. Single cell data like ours involving tens of thousands of cells, thousands of reads per cell, and multiple time points and treatment groups enable a very wide array of analyses. We acknowledge that we have not exhausted the analyses that are possible with these data, but we respectfully disagree that we have been very superficial. First, to identify the cell types we used a variety of approaches to avoid incorrect identification. This is not a trivial effort, and we feel we did this rigorously using Gene Ontology analysis, GREAT analysis, marker gene analysis, and validation from published scRNA-seq data (Fig.1c; Fig.2e,g,h; Fig.3a; Extended Data Fig.3). To be specific, the peaks enriched in vascular endothelial cells were closest to the genes involved in angiogenesis and endothelium development (Extended Data Fig.3b,c). With that, to confirm the cell types, we used marker genes used in other literature. For instance, the marker genes we used such as *Cdh5* and *Pecam1* were widely used to identify endothelial cell lineage (Feng et al., 2019 Front. Cardiovasc. <https://doi.org/10.3389/fcvm.2019.00165>). It showed an enriched peak in the promoter of *Cdh5*, an endothelial cell marker) only in vascular endothelial cell population (Fig.5b and Extended Data Fig.2a). Second, in the interest of preserving reader interest, we focused our more in-depth analyses on cell types that we considered most relevant to the physiologic responses associated with adipose functions. These included cell neighborhood abundance changes of all 12 clusters (Fig.1e, f); and focused identification and analysis of functional beige adipocyte behaviors, contrasted with white adipocytes (Fig.2), including gene regulatory functions that integrated our data with published ChIP-seq data (Fig.3). Our developmental trajectories included adipocytes, which we reported (Fig.4). The data used to prepare this figure does not include adipocyte precursors, for reasons we provided in response to this reviewer’s comment 2. We queried reciprocal interactions between adipocytes and vascular cells (Fig.5), and alterations to genes, motifs, and pathways affecting six immune and inflammatory cell populations. Third, we prioritized identifying functional physiologic outputs, as opposed to further detailed analyses specific cell populations. The shared and unique responses to cold exposure or the beta-3 adrenergic receptor agonist at the pathway or lipid metabolism level (Fig.7) that we report had not been previously resolved. Importantly, it was the single cell analyses we described that motivated the lipid analysis, which revealed unexpected differences between responses to cold or the beta-3 adrenergic receptor agonist. We reiterate our agreement that we have not exhausted the descriptive analyses of individual cell types that are possible with these data, but we consider the analyses we did to be thorough and focused on the functional physiologic outputs that are ultimately served by gene regulatory mechanisms.

7. A primary concern is that the follow-up studies described in Figure 7 are generally unconvincing. If the major difference between cold and CL316,243 treatment was lipogenic gene expression, the manuscript would benefit from qPCR or other validation. Also, these studies have a relatively low n number (n = 3). Overall, it is difficult to interpret the relevance of these follow-up studies have on beige adipocyte physiology.

qPCR and other tests to validate the expression effects of the observed differential gene accessibility, are certainly useful, as the reviewer suggested. To fill this gap, we have added the lipogenic gene expression levels from publicly available snRNA-seq data (Extended Data Fig.5i). It showed the increased expression levels of lipogenic genes after 4 days of cold and CL treatment in adipocytes from inguinal adipose tissue, which is consistent with our findings in our snATAC-seq data.

It is important to note that any validated changes in RNA expression would beg the question as to whether there were changes in protein expression; and any validated changes in protein expression would beg the question as to whether those changes had physiologically functional consequences, as opposed to consequences on RNA or protein expression only. Because we observed many differentially accessible genes, and because their gene ontology categories strongly indicated that any physiologically functional effects of altered expression should be felt at the level of lipid profiles, we directly assayed lipid profiles. We consider this to be more proximal to the physiologic outputs that our study sought to address. The findings we report in Fig.7 are those expected if the differential gene accessibility were physiologically functional. Accordingly, we consider the lipid profiling a more direct validation for functional physiologic effects than RNA or protein measures.

8. Although it can be common for GC/MS analysis of FAs to be represented as percentages rather than absolute values, it is difficult to determine whether these changes are due to relative FA production changes or preferential β -oxidation of some FAs over others. It would help if the authors could include the weight of the adipose tissue next to the data. Additionally, the authors could present the data as a stacked bar graph where all the columns add up to 100%. This would highlight that the data is a percentage rather than an absolute.

We agree that a limitation of our lipidomics analyses is that they are steady state measures of lipids present at the time tissues were taken for analysis. To resolve the interesting question the reviewer raises about FA production changes, or preferential β -oxidation requires metabolic labeling and kinetic analyses, which may be motivated by our studies, but they are well outside the scope of what our experiments were designed to do. The value of the suggested studies is only brought to light by the present studies and should not diminish the value of what we have done.

The weight of the adipose tissue used for lipidomics analysis was 40-50mg, as described in the methods (line 893).

We thank the reviewer for the suggestion to improve the presentation of our results. However, we could not use a stacked bar graph, because it could not represent the subtle differences in the percentages of 21 lipid species. Most lipid species have <1% abundances, which cannot be well represented in a stacked bar graph. However, in an

effort to present the data to highlight that the data is a percentage, we now have presented the data as a bar graph, instead of a box plot.

9. What significance does the different fatty acid compositions have on a functional level? The paper would be substantially stronger if the author could discuss the different fatty acid compositions and related adipocyte functions.

There are many potential functional consequences of changes in FA composition, but anything we might propose is necessarily speculative. Nonetheless, it is reasonable to speculate that adaptation to the cold is an important functional consequence. At the mechanistic level, desaturation and elongation alters fatty acyl composition that can preserve membrane fluidity at lower temperatures (Farkas T, Kitajka K, Fodor E, Csengeri I, Lahdes E, Yeo YK, Krasznai Z, Halver JE. PNAS U S A. 2000 doi: 10.1073/pnas.120157297). As components of triacylglycerols, the molecular form in WAT lipid droplets, unsaturated fatty acids lower melting points than their fully saturated analogues. This is accurate for fatty acids/acyl groups with up to 3 double bonds in poikilothermic animals (Dey I, Buda C, Wiik T, Halver JE, Farkas T. PNAS 1993 doi: 10.1073/pnas.90.16.7498).

We have added more sentences to discuss the potential effects of different lipid composition on β -oxidation rate and melting point (lines 392-408).

10. The authors mention the role of palmitic acid in inflammation in their discussion. In general, the inflammatory response is observed with the free fatty acid version of palmitate, whereas the authors are using gas chromatography to investigate both free and esterified versions of palmitate. This should be either further clarified or removed.

We apologize for the ambiguity and agree that this may be too speculative. We have removed this portion of the sentence.

Minor comments:

- Inguinal Adipose Tissue in Figure 1a is missing an "l"

It was a PDF conversion issue. We have corrected these in the revised manuscript.

- In multiple cases (including Figure 1d), the figure description says individual samples are represented with dots – however, I cannot see those individual samples.

We have corrected these in the revised manuscript.

- The resolution on Figures 1 makes it difficult to discern the legends on Figures 1e and 1f (For example, I cannot tell what the difference between an overlap size of 25, 50, or 75 is). Additionally, it is not clear what plotting the overlap size, or the neighborhood size adds to the interpretation of this graph. Generally, all plots seem to have some labels that appear a bit small to read easily.

The size of labels has been increased. Further explanations have been included.

- I think it is a mistake, but Figure 3d appears primarily black.

It was a PDF conversion issue. Now we have corrected it in the revised manuscript.

- Figure 4 describes 3 modules for understanding beige adipocyte development with individual genes representing each module appearing in Figure 4c. These genes could be labeled with their respective modules for clarity. Also, the trajectory in panel c is misspelled. Figure 6 has many misspellings in the title of the figure. Also, b has an overlapping title on the right.

We labeled module information on top of each TF (Fig.4c); other mis-spellings were due to PDF conversion issues and corrected in the revised manuscript.

- Lines 52-54 – This sentence appears to be missing a word or two and is generally a little confusing. I would recommend splitting it into two sentences for clarity.

We have re-worded.

- In the introduction, I think the authors assume that pharmacologic administration of Adrb3 agonists is more practical in clinical settings than cold exposure. If so, they should specifically state it.

We have removed the portion to avoid confusion. What we meant to state in the original manuscript was based on the previous study Jiang and Berry et al., (2017 eLife) showing that pharmacologicals are more likely to gain acceptance than cold environments.

- Line 100 – there is a 22 that appears randomly (I assume it is a miss-formatted reference?)

It was. We have corrected.

- Lines 110-113 – The authors discussed the top marker genes enriched in white adipocytes and beige adipocytes. They should specify whether these genes were used to determine the identity of these subclusters or whether there was a different method for determining their identity.

This was based on differentially accessible analysis. They are the top enriched genes in white and beige adipocytes, which agree with their cell type annotation.

- Discussion of some of the figures (primarily Fig 2c) is out of the order that they appear.

We have rearranged paragraphs in order to match the figure order in the text (lines 104-127).

- The authors could explain the differences between KEGG and GO to explain why they used these different techniques.

Both group lists of genes involved in the same biological process. However, these two provide a slightly different level of information. GO group genes using an ontology, while KEGG group genes into pathways. In other words, GO provides information on the functions of a set of genes. This finds biological process terms associated with the collection of genes. On the other hand, KEGG pathway can describe mechanisms, interactions and/or phenomena, not simple lists of genes.

Often times, ontology annotations are more flexible and can capture more functional relationships. It was especially more informative for us to predict cell type identities. For example, GO provided terms such as 'B cell activation' and 'T cell activation', whereas KEGG provided terms like 'JAK-STAT signaling pathway' or 'Ras signaling pathway'. In some cases, KEGG pathway analysis was more useful to identify a pathway of our interest. To be specific, in dendritic cells, we found that the genes enriched in dendritic cells at day 0 are relevant to NF-kB pathway.

- Lines 372-373 – if this is an important point, then the authors should include the panel in the main figures and not the supplementary figures.

We have added coverage plot of *Eno1* in Extended Data Fig.8i. Although it is an important gene, we thought it is only one of the glycolytic genes that might be better included in the supplementary figures.

Reviewer #2:

1. The concern is that the rationale for dissecting the distinct mechanisms underlying cold and Adrb3 induced beiging is not well established. The action of these stimuli includes recruitment of new beige adipocytes and activation of existing beige adipocytes. As these stimuli may recruit beige cells from distinct (progenitor) cell types, it is obvious that the chromatin accessibility may be quite different as you may be comparing apples to pears. From the standpoint of activation of existing beige cell, shouldn't cold and Adrb3-agonist similarly activate thermogenesis of beige cells via the Adrb3 receptor and downstream signaling? If so, any difference that was observed may be due to different effects/dosages of endogenous NE (induced by cold) and exogenous Adrb3 agonist. Some discussion should be included to address these issues.

Among the unresolved questions in the field is whether there are shared and/or distinct mechanisms for cold and Adrb3 induced beiging. We agree with the reviewer that there are many potential pathways towards beiging that include existing or progenitors of beige adipocytes. Our study is an early foray into characterizing the beiging phenomenon at a single cell level. We clearly identify both shared and distinct chromatin state changes in mice exposed to cold or the Adrb3-agonist. We infer from this that there are overlapping pathways activated by the two stimuli, but also stimuli-specific responses, and these reflect the beiging mechanisms. There may also be differences in kinetics of the responses as well that are read out as chromatin state differences. We have modified our discussion, as suggested, to include these possibilities.

We also agree with the reviewer's point that agonist dosage and levels of endogenous norepinephrine produced by agonist and cold exposure may alter chromatin states in subtle ways that lead to apparent differences between cold or Adrb3-agonist responses. We have added this in our discussion (lines 426-431). Using single-cell methods to evaluate dose responses may be justified as a follow-up to our study.

The title is also misleading as the one on the PDF file is different from that in the one manuscript form.

We have modified the title.

2. As chromatin accessibility may or may not be directly linked to gene transcription, the currently results can be compared to publicly available snRNA-seq results to pinpoint the specific changes. Discussion on the limitation of sn-ATAC and how the limitation can be overcome by other technique should be included. For example, SNARE-seq can reveal both transcriptome and chromatin accessibility within the same cells.

We thank the reviewer for the suggestion. And we have added the limitation in discussion (lines 432-438). Although we did not measure transcriptome and chromatin accessibility in the same nucleus like SNARE-seq, we made this comparison by using publicly available snRNA-seq from the similar experimental settings and reported the findings in Extended Data Fig.3f,g, and Extended Data Fig.5g,i. This comparison confirmed our cell type annotation and our findings from our snATAC-seq data.

3. The Introduction stated that the object of this study was to explore the difference of chromatin accessibility dynamics in cold or Adrb3 agonist-induced beige adipocytes, but the beige-specific transcription factor activity and chromatin interaction networks (Fig. 3)

were only analyzed between white and beige, the effect of treatments at various timepoints were not considered. Similar comparisons (between treatments) were also missing for the beige adipocyte developmental trajectory (Fig. 4). The only available comparison between cold and *Adrb3* agonist treatment was for lipogenic and glycolytic genes (Fig. 7). However, the authors compared the whole adipocyte population, instead of beige adipocytes. The conclusion would be strengthened by including these missing comparisons.

We included most of these comparisons. Fig.2d shows in a condensed way the differences between treatment groups and across time points. Fig.7a-f show shared and distinct gene accessibility patterns between treatment groups and their GO categories. Extended Data Fig.4 shows cell population changes across time points for the two treatment groups. Extended Data Fig. 6 shows gene accessibility changes in adipocytes across pseudotime trajectories for both treatment groups. Extended Data Fig.7 and Extended Data Fig. 8 show volcano plots with differential gene accessibilities in adipocytes for each time point and both treatment groups.

We thank the reviewer for noting this concern about beige adipocyte comparison between cold and CL. We compared the whole adipocyte population for a few reasons. Firstly, we found that most of variable genes found in beige adipocyte comparisons are still identified to be differentially accessible in the whole adipocyte comparisons. We have added scatter plots to show overall gene accessibility changes after cold and CL treatment in whole adipocyte population as well as beige adipocyte population (Extended Data Fig.8g,h). For example, both plots show that *Ucp1* has high log₂-fold change in all adipocytes as well as beige adipocytes after cold and CL treatment. As beige adipocyte population is smaller than white adipocyte, using the whole adipocytes increase the statistical power. Secondly, while both can capture similar gene sets, we noted that beige adipocyte comparison tended to increase log₂-fold change of some genes in cold beige adipocytes, which may exaggerate some of the gene accessibility changes in cold beige adipocytes compared to CL beige adipocytes. We suspect that this is due to the differences in kinetics of the responses that are read out as chromatin state differences, like we discussed before, which are not the focus of our analysis. Therefore, we used whole adipocyte population for this analysis.

4. The relative efficiency of cold and *Adrb3* treatment used in this study on beiging should be validated by at least morphology of iWAT and immunoblot analysis of browning marker proteins, such as UCP1.

As recommended by the reviewer, we have added the histology images (Extended Data Fig.1). Although we did not perform immunoblots, our data suggest that the treatments induced the accumulation of beige adipocytes and created the expected changes in accessibility at marker genes diagnostic of beiging (Fig. 2d-f; Fig 3a, b; Extended Data Fig. 3a,c,e).

5. Fig.1d: The proportion of beige adipocyte under both treatments did not show significant increase. Can the authors explain this result?

The end points of the treatments (7 days) in cold and CL treated mice respectively show a 4 and 6 fold increase in beige cells relative to controls at room temperature. We quantified 12 cell types across two treatments plus controls for three treatment time points for a total of 84 abundance values. Applying Bonferroni correction for multiple hypothesis testing sets a high bar for statistical significance as Bonferroni correction is very

conservative. However, by applying Milo analysis, we found high log fold change of beige adipocyte after cold or CL treatment (Fig 1e,f).

6. Fig. 1a: No vehicle injection control was set for CL-316,243 injection group.

As the reviewer pointed out, a vehicle injection control for CL is often used, especially, in an experimental setting comparing CL-treated mice and vehicle-treated mice at the same time point. For example, a previous single cell RNA-seq study that we also analyzed in our study, included saline control as a vehicle injection control for CL when the comparisons were made at the same time point (day 4) (Rajbhandari et al. 2019, eLife (10.7554/eLife.49501)). However, in our analysis, we focused on the changes before and after interventions at multiple time points (day1, 3 and 7), rather than having each time point or each treatment control, which otherwise dramatically increase the number of control mice. Similar strategy was used in Rajbhandari et al. (2019 eLife). When they collected data from multiple time points (day 1, 2, and 4) and included cold-treated mice, they also used RT control (not vehicle treated).

Moreover, although we used RT as a control group for CL, we found that similar gene sets were enriched after CL treatment compared to RT in our snATAC-seq data and to a vehicle treated mice in RNA-seq data (Wang et al., 2019 Cell Metab. 10.1016/j.cmet.2019.05.005; RNA-seq data on inguinal adipose tissue from 2-month-old CL-treated (4 days) and vehicle-treated mice). Our side-by-side comparisons of the separate studies are in the figure below.

We confirmed that top 20 genes from our snATAC-seq data all have log2FC > 1 in RNA-seq data. Especially, 10 of them were having log2FC > 2 as shown in the table below.

Top 10 genes in snATAC-seq data (CL vs RT)	Wang et al. 2019 RNA-seq (CL vs Vehicle)	
	Log2FC	p_value
Acly	3.29111575	7.93E-26
Acaca	2.22067881	2.62E-13
Tkt	2.33936011	9.18E-12
A830018L16Rik	2.68573553	1.68E-42
Elovl6	2.88503832	2.29E-18
Ppp1r3b	2.94065192	5.66E-18
Khdrbs3	2.44397373	2.98E-33
Ttc25	3.40899076	2.24E-19
Cidea	3.77527681	3.57E-23
Gnao1	2.94941717	1.07E-24

This list includes lipogenic genes (*Acly*, *Acaca*, *Elovl6*) that we reported to be more accessible after CL treatment compared to RT control. This might suggest that differential

analysis compared to RT control still enables us to capture significantly enriched genes/features by CL treatment.

Reviewer #3:

1. Despite using ATAC-seq chromatin accessibility data, most of the results are presented as genes, not as peaks. It is not clear how the peaks were assigned to genes. As this manuscript is missing with corresponding RNA-seq data, it is not shown and unknown how their snATAC-seq data would be actually reflected in gene expression. The authors should state this limitation and also clarify how the peak-to-gene assignment was performed.

We performed both gene level and peak level accessibility analyses. For gene level accessibility, we summed normalized fragment counts intersecting the gene body and promoter regions (+2kb upstream). Our peak level analyses considered intergenic regions outside these genic intervals. For peak-to-gene assignments, we used closest genes from the differentially accessible peaks. These gene level and peak level accessibility analysis details are provided in the methods section (lines 778-785 and 825-826).

We agree with the reviewer that a limitation of accessibility data is that they do not directly report gene expression outputs, and we acknowledge this in discussion (lines 432-438). In the revised manuscript, we integrated our data with published snRNA-seq data and show that the expression outputs we predict from our accessibility data are found in others' RNA data. Importantly, our response to reviewer 1 point 7 bears repeating here: Changes in gene accessibility beg the questions as to whether there are changes in RNA levels, or protein levels, and most importantly, if these are physiologically functional. Our lipid profiles reveal the physiologic relevance of the chromatin state changes we report.

2. Some statements about potential side effects of CL treatment in the Discussion are somewhat overinterpreting the results. For example, there is lack of data to link the increased MUFA content in adipose tissue to an increase in circulating MUFA.

CL is associated with cardiac pathologies by unknown mechanisms. Because we found that CL also led to changes in genes and lipid species that have been implicated in coronary artery disease, we thought it was appropriate to identify these as mechanisms potentially contributing to the side effects of CL. We specifically present these as only as possibilities. However, the link between increased MUFA and circulating MUFA may be too speculative. We have removed this portion of the sentence and discussed a potential effect of CL in regulating cardiac function in discussion (lines 413-431).

3. Some of the Extended Data are not discussed in the text (e.g. Ext Data Fig. 4d-f). All the included data should be discussed or otherwise removed.

Extended Data that are not discussed have been either removed or discussed.

4. In Fig 4b & c, the label for pseudotime is confusing. If pseudotime starts from 0 to 100, it is interpreted as white adipocytes (colored blue) appearing from beige adipocytes (colored yellow), which is contrary to the color key in Fig 4b.

We thank the reviewer for pointing out this. We have corrected it in Fig.4b and changed the colors overall to be more consistent across the figures.

Some typos and minor errors:

- Line 14: though -> through

- Line 212: peudotime -> pseudotime
- Fig 3d: black background
- p33, Fig 1d, there is no dot, while the legend states “Dots represent each biological replicate”.
- Fig 1a: nguinal -> ingunal
- p34, Fig 2d labeling

Most of these errors were due to a PDF conversion issue. We have corrected it in the revised manuscript.

Updated figures:

Overall, all figures are resized and colors are changed to be more consistent.

- Fig.1a – changed due to a pdf conversion error
- Fig.1b – colors changed
- Fig.1d – colors changed
- Fig.3d – changed due to a pdf conversion error
- Fig.3g – colors changed
- Fig.4a – resized
- Fig.4b – colors changed
- Fig.4c – module information is included
- Fig.4e – rearranged
- Fig.4h – a graphical summary is added
- Fig.5a – colors changed
- Fig.5d – colors changed
- Fig.7a,b – heatmap is replaced with venn diagram to show overlaps among days and between treatment conditions
- Fig.7e – GO for genes enriched in both treatment conditions is included
- Fig.7f – a line plot is replaced with a box&violin plot due to the discontinuous nature of the data
- Fig.7g-j – colors changed
- Fig.7k – a graphical summary is added
- Extended Data Fig.1 – H&E stained images is added as requested
- Extended Data Fig.2c – colors changed
- Extended Data Fig.3 – rearranged
- Extended Data Fig.3a – colors changed
- Extended Data Fig.5 – rearranged
- Extended Data Fig.5a,f,h – colors changed
- Extended Data Fig.7 – rearranged and separated into Extended Data Fig.7 and 8
- Extended Data Fig.8g-i – scatter plots of genes are included to show the overall distribution of log fold change of gene accessibility after cold and CL treatment compared to RT day 0 control. And coverage plot of *Eno1* is included in addition to gene accessibility of *Eno1*.
- Extended Data Fig.9 – colors changed

REVIEWERS' COMMENTS:

Reviewer #1 (Remarks to the Author):

The authors have adequately addressed all the previous concerns.

Reviewer #2 (Remarks to the Author):

The authors did a good job addressing my previous concerns and the manuscript is much improved. My only suggestion at this point is for the authors to outline (explicitly explain) in introduction or in the discussion the main physiological differences between cold and Adrb3 agonist treatments. My understanding is that cold stimulates sympathetic nervous system that eventually activates Adrb3 on brown and beige adipocytes. If that is the case, then the two stimuli largely overlap. Are there any additional targets of cold treatment that is not mimicked by Adrb3 agonist? Likewise, are there any other off-target effects of CL beyond the brown/beige adipocytes and Adrb3 receptor? I believe that explaining the physiological context would also better highlight the significance of the current study.

Reviewer #3 (Remarks to the Author):

The authors have successfully addressed all of my comments except for one thing (point #1). The authors say that the details of their peak- and gene-level analyses are provided in the Methods section but I could not find them. Please make sure that the information is correctly included in the manuscript.

COMMSBIO-21-3089 Response to Reviews

We are grateful for the positive reviews, and are encouraged by the reviewers' and the editor's interest in our work. Below in italics we provide responses to the remaining reviewer's points highlighted in red. The revised manuscript includes changes that we made in response to reviewer #2 (lines 49-53) and as the editor requested in the Editorial Requests Table.

Reviewer #1:

The authors have adequately addressed all the previous concerns.

Reviewer #2:

The authors did a good job addressing my previous concerns and the manuscript is much improved. **My only suggestion at this point is for the authors to outline (explicitly explain) in introduction or in the discussion the main physiological differences between cold and Adrb3 agonist treatments.** My understanding is that cold stimulates sympathetic nervous system that eventually activates Adrb3 on brown and beige adipocytes. If that is the case, then the two stimuli largely overlap. Are there any additional targets of cold treatment that is not mimicked by Adrb3 agonist? Likewise, are there any other off-target effects of CL beyond the brown/beige adipocytes and Adrb3 receptor? I believe that explaining the physiological context would also better highlight the significance of the current study.

We sincerely appreciate the reviewer's suggestion. The major point raised by the reviewer has been addressed in introduction (lines 49-53).

A non-canonical mechanism by cold exposure was suggested by Dr. Kajimura group (Chen et al., Nature 2019 doi:10.1038/s41586-018-0801-z). They showed that beige adipocytes formed by cold exposure in the absence of β -adrenergic receptor signaling have enhanced glycolytic activities. We also reported relatively increased gene accessibilities involved in glycolysis in beige adipocytes induced by cold exposure compared to the beige adipocytes induced by CL treatment (lines 385-389). We think this may be an extra target of cold exposure that is not mimicked by CL.

Adrb3 is expressed in several tissues but most highly expressed in adipocytes. The known off-target effects of CL include tachycardia and hypertension (lines 58-59) and we discussed the potential effects of CL in cardiac contraction in discussion (lines 424-433).

Reviewer #3:

The authors have successfully addressed all of my comments except for one thing (point #1). The authors say that the **details of their peak- and gene-level analyses are provided in the Methods section but I could not find them. Please make sure that the information is correctly included in the manuscript.**

We apologize for referencing the wrong lines. They are included in lines 770-777 and 815-818 in methods.